# Adaptation to cystine limitation stress confers a targetable lipid metabolism vulnerability in pancreatic ductal adenocarcinoma

Yunzhan Li[1,2,14], Zekun Li[3,4,14], Qin Li[1,2], Dongxiao Sun[5,6], Bo Ni[3,4], Mingjun Tan [1,2], Ashley E. Shay [7,8], Min Wang[1,2], Chenyang Meng[3,4], Guangcong Shen[3,4], Boyang Fu[3,4], Yueying Shan[3,4], Shiqi Zhang[9], Rifah Rownak Tanshee [1,2], Tianxing Zhou[3,4], Yongjie Xie[3,4], Kun-Ming Chen[2,10], Bin Qiao[11], Yunkun Dang[12], Scot R. Kimball [1], Guanshi Zhang [9], Girish H. Rajacharya[13], Pankaj K. Singh [13], Xiuchao Wang [3,4] ✉, Jihui Hao [3,4] ✉ & Shengyu Yang [1,2] ✉

Cystine/cysteine is critical for antioxidant response and sulfur metabolism in cancer cells and is one of the most depleted amino acids in the microenvironment of pancreatic ductal adenocarcinoma (PDAC). The effects of cystine limitation stress (CLS) on PDAC progression are poorly understood. Here we report that adaptation to CLS (CLSA) promotes PDAC cell proliferation and tumor growth through translational upregulation of the oxidative pentose phosphate pathway (OxPPP). OxPPP activates the de novo synthesis of nucleotides and fatty acids to support tumor growth. On the other hand, CLSA-mediated lipidomic reprogramming depends on triacylglycerides synthesis and lipid droplet formation to mitigate lipotoxicity. Through drug screening, we identify lomitapide as an inhibitor of CLSA PDAC tumor growth and a potent sensitizer of chemotherapy. Lomitapide inhibits triacylglycerides synthesis to interfere with CLSA and chemotherapy-induced lipidomic reprogramming. Taken together, we demonstrate that CLSA promotes PDAC tumor growth through metabolic reprogramming and lomitapide could be used to target the dysregulated lipid metabolism in PDAC.

Pancreatic ductal adenocarcinoma (PDAC) is a highly aggressive cancer with a 5-year survival rate less than 13%. The PDAC tumor microenvironment is characteristically nutrient deprived due to desmoplasia, poor vascularization, and hyperactive cancer metabolism[1,2]. Plasticity in cancer metabolism is critical for cancer cells to overcome nutrient limitation stress and to support tumor growth and progression[1]. However, the roles of nutrient limitation stress in PDAC tumor growth and progression remained incompletely understood.

Cystine ($Cys_2$) and cysteine (Cys) are important amino acid nutrients for the antioxidant response and sulfur metabolism[3,4]. In the oxidative environment in the blood, Cys is oxidized into $Cys_2$. SLC7A11, the functional subunit of the $Cys_2$/glutamate antiporter system xc⁻, is responsible for cystine uptake[3,5]. In the cell, $Cys_2$ is reduced to Cys and used for the synthesis of essential sulfur metabolites, such as glutathione (GSH) and iron-sulfur clusters (Fe-S)[6]. Although $Cys_2$ and Cys are non-essential amino acids, cancer cells depend on exogenous $Cys_2$ for antioxidant response and survival[4,7]. $Cys_2$ deprivation induces

ferroptosis, a type of iron and lipid peroxidation-dependent cell death in pancreatic cancer and other cancers[7–12]. Intriguingly, $Cys_2$ and Cys are among the most depleted amino acids in the PDAC tissue and tumor interstitial fluid[13,14]. It is unclear how PDAC cells adapt to the chronic $Cys_2$ limitation stress (CLS) and how such adaptation affects PDAC tumor growth and progression.

Cancer cells upregulate anabolic metabolism, such as de novo synthesis of nucleotides and fatty acids (FA), to support cancer cell proliferation and tumor growth[15]. The oxidative pentose phosphate pathway (OxPPP) provides ribose-5-phosphate (R5P) for the de novo synthesis of nucleotides[16] and NADPH for de novo FA synthesis[17,18]. glucose-6-phosphate dehydrogenase (G6PD), 6-phosphogluconolactonase (PGLS), and phosphogluconate dehydrogenase (PGD) are the three enzymes in the OxPPP, which use glucose-6-phosphate (G6P) as precursor for the synthesis of R5P and the reduction of NADP+ to NADPH[16]. G6PD catalyzes the rate limiting step of OxPPP and is upregulated in many cancer types, including PDAC[19].

While the upregulation of lipogenesis provides lipids to support membrane synthesis during cancer cell proliferation, the accumulation of excess FA could lead to lipotoxicity and cell damage[20–22]. To protect cells from lipotoxicity, excess FA is esterified by diacylglycerol O-acyltransferase 1 and 2 (DGAT1 and 2) into triacylglycerides (TG) and stored in the lipid droplets (LD)[22,23]. LD accumulation correlates with tumor aggressiveness and therapy resistance in many cancer types[24]. In addition to serving as lipid storage organelles, LDs are crucial for cell survival under stress. Stress factors such as ER stress, acidosis, and chemotherapeutic agents can promote lipolysis and/or lipogenesis[23,25]. The storage of excess FA, polyunsaturated fatty acids, lipid peroxidation products, and acyl-ceramides in LD allows cells to avoid stress-induced lipotoxicity and cell death[26]. There are recent reports indicating that DGAT1/2 inhibitors could be used to inhibit tumor growth and improve the efficacy of chemotherapy and radiotherapy in colorectal cancer, melanoma, and glioblastoma[20,27]. However, the roles of LD in PDAC remain poorly understood[28].

Here, we investigated the effects of chronic CLS in PDAC. We found that adaptation to CLS (CLSA) promotes PDAC cell proliferation, ferroptosis resistance, and tumor growth. CLSA PDAC cells translationally upregulate OxPPP enzymes to promote biosynthesis of NADPH, nucleotides, and FAs. Through screening of a metabolism-focus drug library, we identified that lomitapide could potently induce lipotoxicity in CLSA PDAC by suppressing TG synthesis, LD formation, and increasing toxic lipid species. Our data further showed that the FOLFIRINOX chemotherapy regimen induces lipidomics reprogramming and lomitapide could target FOLFIRINOX-induced LD formation to increase therapeutic efficacy. In patient-derived xenograft models and the $Kras^{LSL-Kras\ G12D/+};\ Trp53^{LSL-\ R172H/+};\ Pdx1-Cre$ (KPC) genetically engineered mouse model, lomitapide inhibited PDAC tumor growth and robustly improved the efficacy of FOLFIRINOX regimens. Our data shed new light on the role of nutrient stress in PDAC progression and showed that lomitapide could be used to exploit stress-induced vulnerability in PDAC lipid metabolism.

## Results

### Adaptation to cystine limitation stress promotes PDAC cell proliferation, ferroptosis resistance, and tumor growth

To determine whether PDAC cells might be subjected to CLS, we determined the levels of Cys and $Cys_2$ in PDAC and para-tumor normal pancreatic tumor tissue from C57BL/6 mice orthotopically implanted with a PDAC line derived from the KPC ($Kras^{LSL-G12D};Trp53^{LSL-R172H};\ Pdx1-Cre$) model. The levels of Cys and $Cys_2$ in tumor tissues were approximately 90% (for Cys) to 50% (for $Cys_2$) lower when compared to para-tumor normal pancreatic tissues (Fig. 1A). Similarly, we observed 60% and 75% decrease in the levels Cys and $Cys_2$, respectively, in patient PDAC tissues when compared to para-tumor normal

pancreas control in a cohort of 11 PDAC patients (Fig. 1B). We further used Mass spectrometry imaging (MS imaging) to compare the spatial distribution of Cys and $Cys_2$ in tumor tissues from the KFC ($Kras^{LSL-G12D};Trp53^{fl/fl};\ Pdx1-Cre$) spontaneous PDAC model. Our data showed that Cys levels and Cys/Cys2 ratios were decreased in pancreatic intraepithelial neoplasia (PanIN) and further decreased in carcinoma regions when compared to normal pancreatic tissues (Figs. S1A and 1C). Taken together, our data suggest that PDAC cells are subjected to chronic CLS during tumorigenesis and progression.

Most cell culture media (e.g., DMEM or RPMI) contain 200 μM $Cys_2$, which is significantly higher than $Cys_2$ levels in the human plasma (25–50 μM)[29]. Cells cultured in vitro have higher demand for $Cys_2$ due to direct exposure to ambient $O_2$. As shown in Fig. S1B, decreasing $Cys_2$ concentrations to 50 μM or lower in MiaPaCa-2 media led to increased phosphorylation of eIF2α and the upregulation of ATF4 protein levels, which suggested the induction of the integrated stress response by CLS. As expected, acute reduction of $Cys_2$ from 200 μM to 50 or 25 μM led to lipid peroxidation and ferroptosis-like cell death in PDAC cells in 72 h (Fig. S1C–E). In colony formation assay, $Cys_2$ limitation inhibited colony formation (Fig. S1F). Approximately 5 to 30 percent of PDAC cells were able to adapt to CLS at 25 μM and resume cell growth and proliferation after 3–4 weeks exposure to CLS (Figs. S1F, G and 1D, E). Supplementation with 5% BSA in $Cys_2$ limited media (25 μM $Cys_2$) not only had no rescue effect, but paradoxically further inhibited colony formation (Fig. S1F). The proportion of PDAC cells adapted to 50 μM CLS was much higher (Fig. S1F). Here we use CLSA-25 and CLSA-50 to refer to PDAC cells adapted to 50 or 25 μM $Cys_2$ limitation stress, respectively. Culturing PDAC cells in CLS-25 media (25 μM $Cys_2$) reduced the intracellular Cys levels by approximately 75% to 90% in 24–48 h (Fig. 1F). Although Cys level modestly recovered in CLSA-25 MiaPaCa-2 and PANC-1 cells after 4-week adaptation, it was still ~50–60% lower than naïve control cells. No recovery of intracellular Cys was observed in CLSA-25 Pan02 and KPC cells (Fig. 1F). Interestingly, CLSA PDAC cells proliferate much faster than parental cells, despite severe reduction of intracellular Cys (Figs. 1G and S1H). After 4-week adaptation, the ATF4 levels in CLSA cells were no longer elevated, which indicated that CLSA cells overcame integrate stress response despite lower intracellular Cys (Fig. S1I). CLSA-25 cells had increased ability to uptake FITC-$Cys_2$ (Fig. S1J). Consistent with this observation, when cultured in $Cys_2$ replete media, the intracellular Cys levels in CLSA cells were much higher than control cells (Fig. S1K). However, there was no increase in SLC7A11 abundance in CLSA cells (Fig. S1I). Doxycycline-induced chronic knockdown of SLC7A11 (for 4 weeks) also increased cell proliferation (Fig. 1H). Given the considerable interest in using $Cys_2$ deprivation agents as potential cancer therapeutics, we asked whether CLS induced by pharmacological SLC7A11 inhibitors might have similar pro-proliferation effects on PDAC cells. We previously reported that imidazole ketone erastin (IKE) induced ferroptosis in PDAC cell lines and patient-derived organoids (PDOs)[30]. As shown in Fig. 1I, PDAC organoids adapted to IKE-induced CLS (CLSA-I, for inhibitor induced CLSA) also grew faster than naïve control PDOs.

GPX4 is the sole GSH peroxidase responsible for scavenging lipid peroxide and preventing ferroptosis. As shown in Fig. 1J, K, the $IC_{50}$ of RSL-3 in CLSA-25 cells and CLSA-I PDOs increased dramatically by ten to 500-fold when compared to their respective naïve controls. CLSA-50 cells and SLC7A11 chronic knockdown cells also developed RSL-3-induced ferroptosis resistance, although to a lesser extent when compared to CLSA-25 cells (Figs. 1J and S1L). In addition to the GSH-GPX4 axis, $CoQ_{10}$ and $BH_4$ have been reported to be potent suppressors of lipid peroxidation and ferroptosis[31–34]. Interestingly, CLSA PDAC cell were more sensitive to FSP1 inhibitor (iFSP1) and DHFR inhibitor (methotrexate) (Fig. S1M, N), which indicated increased reliance of these alternative anti-ferroptosis pathways in CLSA PDAC cells.

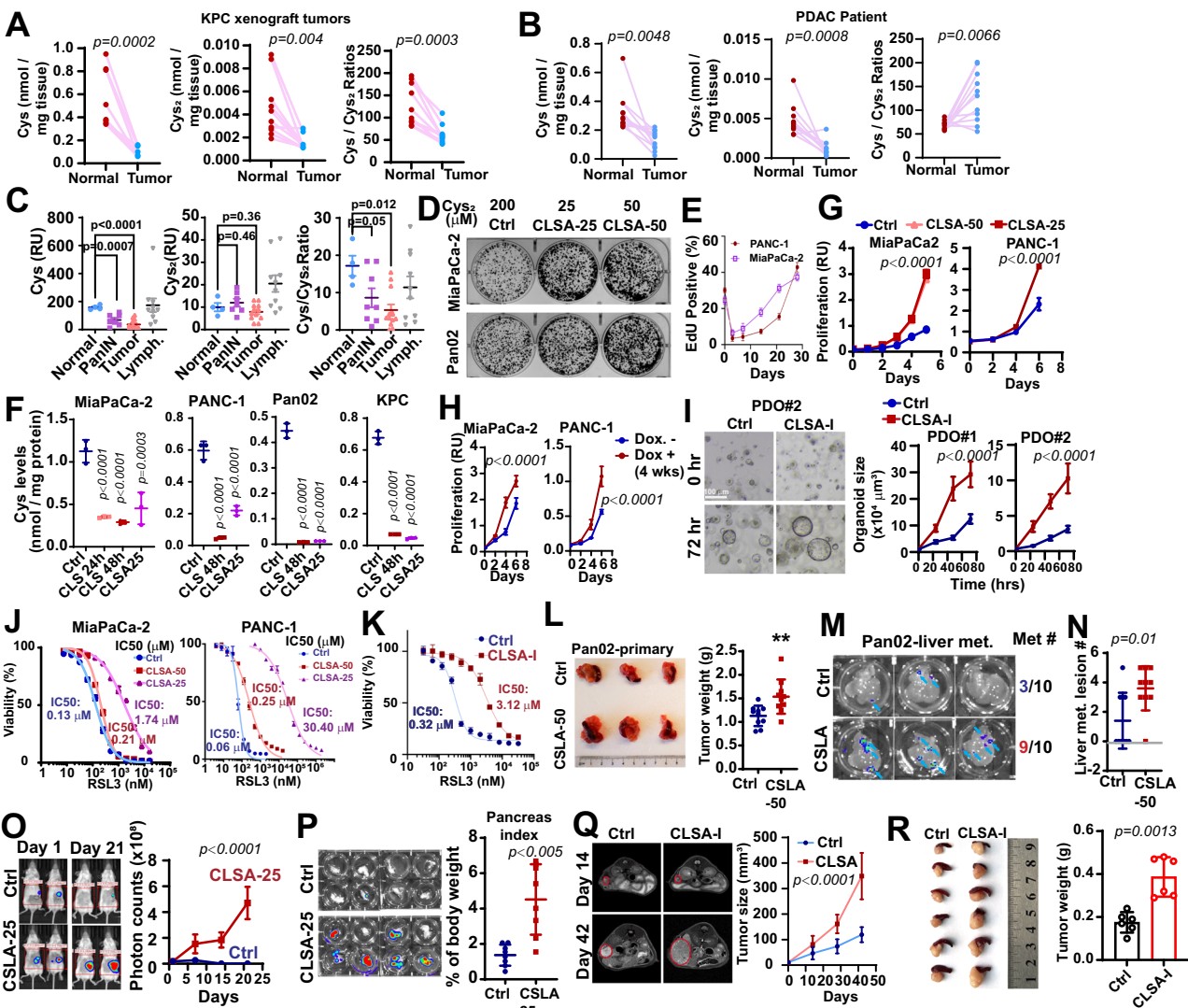

**Fig. 1 | Adaptation to Cys₂ limitation stress promotes PDAC cell proliferation and tumor growth. A**, **B** The levels of Cys, Cys₂, and Cys/ Cys₂ ratios in PDAC tissues (Tumor) and paired para-tumor normal (Normal) pancreatic tissues from KPC xenograft mice ($n = 10$) (**A**) and PDAC patients ($n = 11$) (**B**) were determined through LC-MS/MS after NEM derivatization. **C** Relative Cys, Cys₂ levels, and Cys/ Cys₂ ratios in para-tumor pancreatic tissues (Normal), PanIN, PDAC (Tumor), and tumor-infiltrating lymphocytes (Lymph.) were determined through MALDI-MS imaging of frozen sections of 8 weeks-old KFC mice. **D** The effect of CLSA on colony formation assay MiaPaCa-2 and Pan02 cells. CLSA-25 and CLSA-50 cells were cultured in Cys₂-limited media containing 25 µM and 50 µM Cys₂, respectively. Parental cells cultured in Cys₂ replete (200 µM) medium were used as control. **E** Parental MiaPaCa-2 and PANC1 cells were cultured in 25 µM Cys₂-limited media for 3–28 days and the effects of CLS on cell proliferation were determined through EdU staining and flow cytometry. **F** Cys levels in PDAC cells after being cultured in Cys₂-limited (25 µM) media for 24–48 h, or after adaptation to 25 µM Cys₂-limitation for 4 weeks were determined through LC-MS/MS after NEM-derivatization. **G** The effects of CLSA on cell proliferation in PDAC cell lines. **H** The effects of chronic doxycycline-induced knockdown of SLC7A11 (4 weeks) on cell proliferation in MiaPaCa-2 and PANC-1 cells. Data are shown as mean ± SEM from 4, 7, 12, and 10 ROIs for Normal, PanIN Tumor, and Lymphocytes (Lymph.), respectively. **I** Representative bright-field microscopy (left) and quantitation (right) of tumoroid size showing the effects of CLSA-I on organoid growth in two PDAC PDO models. **J**, **K** The effect of CLSA-25, CLSA-50 (**J**), and CLSA-I (**K**) on RSL3-induced ferroptosis in MiaPaCa-2, PANC-1 (**J**), and PDO1 (**K**). $n = 3$. **L** Representative image (left) and quantitation of tumor weight (right) of orthotopic xenograft tumors harvested from C57BL/6 mice implanted with control or CLSA-50 Pan02 cells ($n = 10$). **M** ex vivo BLI imaging showing the liver metastatic lesions from mice in (**L**). Right, fraction of mice that developed liver metastasis in control and CLSA-50 group. **N** Quantitation of liver metastatic lesion numbers from mice in (**L**) ($n = 10$). **O** Left, Representative BLI imaging from Albion BL6 mice implanted with control of CLSA-25 KPC tumor cells on day 1 and day 21. Right, quantitation of BLI imaging (photon counts) over time. ($n = 10$). **P** ex vivo BLI imaging (left) and the weight (right) of pancreas harvested from (**O**) ($n = 10$). **Q** Representative MRI images (left) and quantitation of tumor volume of mice implanted with control or CLSA-I PDO1. ($n = 6$ per group). **R** Photos (left) and weight (right) of pancreas ($n = 6$) harvested from mice in Q. Data in (**A**) and (**B**) were analyzed using two-sample, paired *t*-test; data in (**C**, **L**, **N**, **P**, and **R**) were analyzed using two-sample unpaired *t*-test and not adjusted for multiple comparison. Data in (**G**–**I**, **O**, and **Q**) were analyzed using two-way ANOVA. Data in (**F**) were analyzed using one-way ANOVA followed by Dunnett's multiple comparison test. Data in (**E**–**G**) and (**H**–**K**) are representative results from at least three independent experiments. All error bars are mean ± SD except for (**C**) and (**I**), which are shown as mean ± SEM. All statistical tests are two-sided.

To understand the potential mechanisms underlying CLSA-mediated ferroptosis resistance, we investigated the protein levels of major regulators of ferroptosis resistance in GPX4 and CoQ₁₀ axis, including GPX4, FSP1, DHODH, COQ2, COQ6, and COQ9. As shown in Fig. S1O, P, there was little change in GPX4, COQ6, and COQ9 levels.

However, FSP1 levels were downregulated, while COQ2 was modestly upregulated in CLSA PDAC cells. To determine the role of the CoQ₁₀ pathway, we used iFSP1, brequinar (BQR), and COQ2 shRNA to suppress the ubiquinol recycling (through FSP1 and DHODH inhibition) and CoQ₁₀ synthesis (through COQ2 knockdown). COQ2 knockdown

almost completely, and iFSP1 or BQR treatment partially, overcame RSL-3-induced ferroptosis resistance in CLSA PDAC cells (Fig. S1Q). Taken together, our data indicated that the ferroptosis resistance in CLSA PDAC cells was at least partially dependent on the $CoQ_{10}$ anti-ferroptosis axis.

To determine whether CLSA-mediated PDAC cell proliferation and ferroptosis resistance depends on continuous CLS pressure, we cultured CLSA MiaPaCa-2 cells in $Cys_2$ replete (200 μM) media for 4 weeks. As shown in Fig. S1R, S, even after 4 weeks culture in $Cys_2$ replete conditions, the CLSA cells maintain elevated cell proliferation and ferroptosis resistance. These data indicated that CLSA phenotypes, once established, are relatively stable.

To determine whether CLSA affects tumor growth, we implanted CLSA Pan02 and KPC cells and their naïve control cells orthotopically into the pancreas of C57BL/6 mice. Mice implanted with CLSA-50 Pan02 cells grew modestly larger tumors, with increased numbers of liver metastases when compared to the control group (Fig. 1L–N). The liver metastasis incident increased from 30% in control to 90% in CLSA-50 group (Fig. 1M, N). In the KPC xenograft experiment, the growth of orthotopically implanted CLSA-25 KPC tumors was dramatically faster than control tumors, although no metastasis was detected in either group (Fig. 1O, P). Similarly, orthotopically implanted CLSA-I PDOs developed larger tumors when compared to naïve control (Fig. 1Q, R). Taken together, our data showed that CLSA promotes PDAC cell proliferation, ferroptosis resistance, and tumor growth.

To determine whether nutrient limitation of other amino acids might induce adaptations like CLSA, we cultured MiaPaCa-2 cells in media containing reduced amounts of glutamine (400 μM), tryptophan (10 μM), or arginine (2 μM), respectively. These amino acids and concentrations were selected based on previous reports showing their depletion in PDAC[13,35]. MiaPaCa-2 cells were able to proliferate in Trp-limited media, but not in Glu- or Arg-limited conditions for 2 weeks (Fig. S1T). BSA supplementation allowed PDAC cell adaptation to glutamine limitation (Fig. S1U), as previously reported[14]. However, Glu-limitation adapted cells (in the presence of 5% BSA) proliferate slower than naïve control (Fig. S1V). Although MiaPaCa-2 cells could proliferate under Trp limitation conditions for 2 weeks (Fig. S1T), most cells stopped proliferating after 2 weeks (Fig. S1V, W), which could not be rescued by BSA supplementation (Fig. S1W). Therefore, our data suggested that CLS induced a unique adaptation mechanism in PDAC to promote cell proliferation and tumor growth.

## CLSA cells downregulate iron sulfur cluster synthesis to promote glycolysis

A major function of Cys is to be used for GSH synthesis, which is required for antioxidant response and ferroptosis resistance. Acute $Cys_2$ limitation (25 μM) markedly reduced GSH levels by fifty to ninety percent in PDAC cells (Fig. S2A), which is consistent with the induction of ferroptosis by acute CLS. However, GSH levels in CLSA-25 PDAC cells were comparable to their naïve control cells (Fig. S2A), despite little to no recovery in intracellular Cys. To understand the effects of CLSA on Cys metabolism, we used $^{15}N_2$, $^{13}C_6$-cystine to label Cys metabolites in MiaPaCa-2 cells. Control or CLSA-25 cells were incubated with 200 μM (for control) or 25 μM (for CLSA-25) $^{15}N_2$, $^{13}C_6$- $Cys_2$ for 0, 2, or 4 h. The fractional labeling of M + 4 Cys reached 68% at 2 h and plateau after that in the control group (Fig. 2A). The M + 4 Cys fractions in CLSA-25 group were 54% and 38% at 2 and 4 h, respectively. Despite lower fractional labeling of M + 4 Cys in CLSA-25 cells, the fractional labeling of GSH and GSSG levels in the two groups were similar (Fig. 2B), which suggested faster turnover of GSH/GSSG in CLSA-25 cells.

We noted that the labeling of M + 4 Alanine was reduced in CLSA-25 cells by 80% to 90% after 4 or 2 h labeling, respectively (Fig. 2C). Alanine is a by-product of Cys desulfurase NFS1, which uses Cys as substrate for the synthesis of Fe-S[36]. Fe-S is a critical co-factor for oxidoreductases such as aconitase (ACO) and mitochondrial

respiratory complexes I, II and III[36]. We confirmed that the activities of ACO1 and ACO2 were markedly reduced in CLSA-25 PDAC cells (Fig. 2D). The protein levels of NDUFS1, a Fe-S binding subunit in complex I, was downregulated in all the CLSA-25 PDAC cells. The downregulation of Fe-S proteins in complex II (SDHB) and complex III (RISKE) was also observed in CLSA MiaPaCa-2 cells (Fig. 2E). The downregulation of respiratory complex proteins was not due to global suppression of mitochondria biogenesis, since there was no significant difference in mitochondria mass between control and CLSA cells (Fig. S2B). The upregulation of TFRC, IRP2 in CLSA cells was consistent with the notion that downregulation of the Fe-S synthesis activates Iron Starvation Response (Fig. 2F). Interestingly, ISCU and NFS1, two core components of the iron-sulfur cluster assembly complex, were significantly downregulated in CLSA PDAC cells (Fig. 2F). Mitochondria stress test and glycolysis stress test showed that CLSA cells markedly downregulated mitochondrial oxygen consumption rates (OCR) while upregulating extracellular acidification rate (ECAR) (Fig. 2G, H). Glucose uptake was also dramatically upregulated in CLSA cells (Fig. S2C). Our data suggested that by downregulating Fe-S synthesis, CLSA cells promote a switch from mitochondrial oxidative phosphorylation to glycolytic metabolism. The suppression of Fe-S assembly could be due to the decreased abundance of iron-sulfur assembly complex, the lower intracellular Cys, or the combination of both.

Interestingly, the steady-state labeling of Cys, GSH, GSSG, and CoA all decreased in CLSA-25 cells after 24 h incubation with $^{15}N_2$, $^{13}C_6$-$Cys_2$ (Fig. S2D), which suggested that CLSA-25 cells might enhance their ability to obtain Cys from alternative sources. Since CSE was upregulated in CLSA MiaPaCa-2 (Fig. S2E), we used $^{13}C_1$-Serine as tracer to investigate the effects of CLSA on de novo Cys synthesis through the transsulfuration pathway. Despite approximately 30% labeling of cystathionine by $^{13}C_1$-Serine, the M + 1 Cys was less than 1% in naïve and undetectable in CLSA MiaPaCa-2 cells (Fig. S2F). Although there was 20-30% labeling of M + 1 GSH (Fig. S2F), this was likely due to conversion of $^{13}C_1$-Serine to $^{13}C_1$-Glycine, which is subsequently used for GSH synthesis. These observations are similar to findings reported by the DeNicola's group[37] and suggest that transsulfuration pathway is unlikely a major source of Cys in either naïve or CLSA cells.

It has been previously reported that cancer cells could acquire amino acids through micropinocytosis[14,38]. CLSA PDAC cells significantly increased micropinocytosis uptake of FITC-dextran, which could be blocked by EIPA (Fig. S2G, H). Therefore, micropinocytosis of extracellular proteins could potentially serve as an alternative source of Cys in CLSA PDAC cells.

## Metabolic reprogramming in CLSA cells activates PPP metabolism and nucleotide synthesis

To understand the CLSA-induced metabolic reprogramming, we performed metabolomics screening using control and CLSA-25 PDAC cells (Figs. 3A and S3A–D). Metabolic set enrichment analysis (MSEA) showed that Warburg effects, Glycolysis, Pentose phosphate pathway (PPP), and nucleotide metabolisms (both Purine and Pyrimidine) were among the most overrepresented metabolic pathways in CLSA-25 MiaPaCa-2 and/or PANC-1 cells (Fig. 3A).

Nucleotides are critical building blocks required for the synthesis of rRNA, mRNA, and DNA in proliferating cells[15]. PPP is a glycolysis shunt pathway that provides R5P and PRPP (Phosphoribosyl pyrophosphate), which are essential precursors for the synthesis of both purine and pyrimidine nucleotides[15,39]. CLSA-25 cells have increased levels of multiple ribonucleotides and deoxyribonucleotides (Figs. 3B and S3E, F). To determine whether CLSA activates PPP and de novo nucleotide synthesis, we used [U-$^{13}C$]-glucose and LC-MS/MS to analyze the effects of CLSA on the glucose entry into glycolysis and PPP metabolism (Fig. 3C–E). As shown in Fig. 3D, E, CLSA increased glucose flux into glycolytic metabolites (Fig. 3E), PPP metabolites (R5P, PRPP), and nucleotides (IMP, UMP) (Fig. 3E). Our data suggested that CLSA-

 

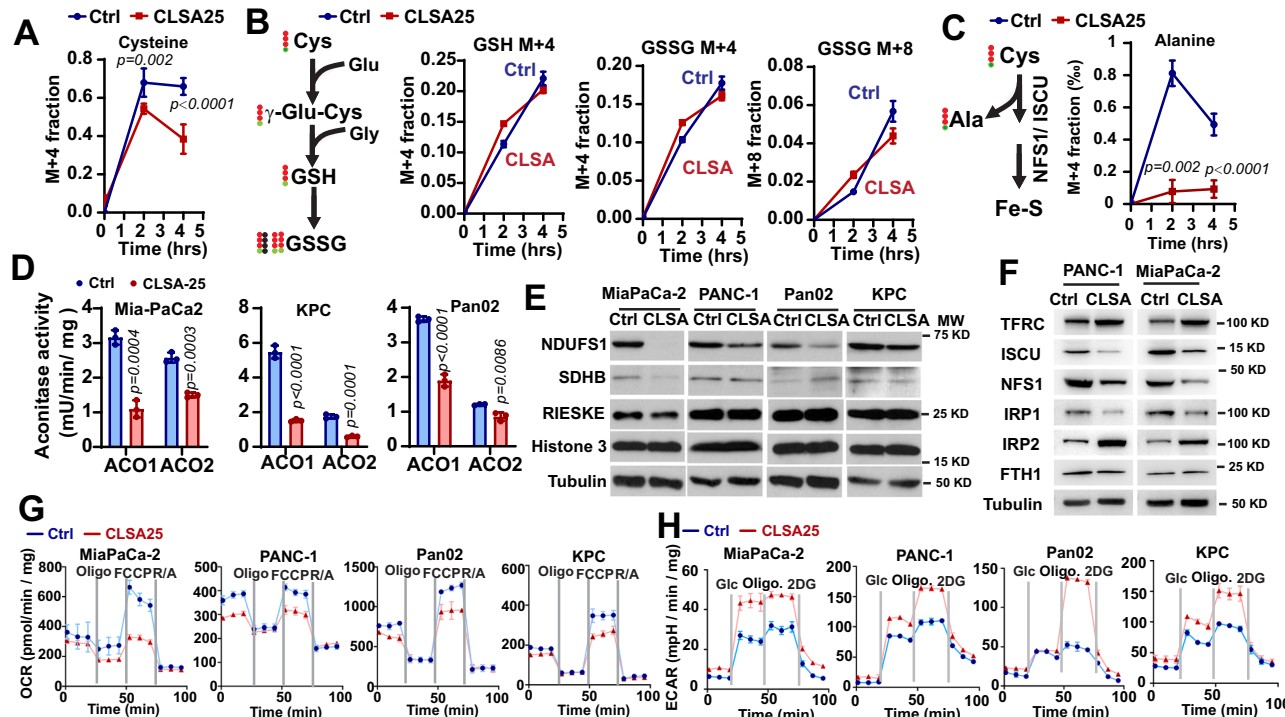

**Fig. 2 | CLSA cells downregulate Fe-S synthesis to promote glycolysis.**
**A**–**C** Fractional labeling of Cys (**A**), GSH, GSSG (**B**), and Ala (**C**) in control and CLSA-25 MiaPaCa-2 cells after incubation with 200 μM (in control group) or 25 μM [$^{13}C_6$, $^{15}N_2$]-Cys$_2$ for 2 or 4 h. **D** The effect of CLSA-25 on aconitase 1 (ACO1) and 2 (ACO2) activities in PDAC cells ($n = 3$ biological replicates). **E** Western blotting analysis showing the proteins levels of Fe-S proteins in mitochondrial complex I (NDUFS1), II (SDHB), and III (RIESKE) in control or CLSA-25 PDAC cells. **F** Western blotting analysis showing the levels of proteins involved in iron-starvation response (TFRC, FTH1, IRP1, and IRP2) and Fe-S assembly (ISCU and NFS1) in control or CLSA PANC1 and

MiaPaCa-2. **G**, **H** The effects of CLSA-25 on mitochondrial oxygen consumption rate (OCR) (**G**) or extracellular acidification rate (ECAR) (**H**) in MiaPaCa-2, PANC-1, Pan02, and KPC cells. $n = 4$ biological replicates. Western blotting samples in (**E**) and (**F**) were derived from the same experiment, but different gels for different antibodies and processed in parallel. Data in (**A**–**D**) were analyzed using two-sample, two-tailed, unpaired $t$-test. Data in (**D**–**H**) are representative results from at least three independent experiments ($n = 3$ biological replicates per group for each experiment). Data in (**A**–**C**) are results from one experiment. All error bars are as mean ± SD.

mediated metabolic reprogramming promotes glycolysis, PPP, and de novo nucleotide synthesis to support cancer cell proliferation and tumor growth.

## OxPPP is translationally upregulated in CLSA PDAC cells to promote nucleotides and NADPH synthesis

G6PD and transketolase (TKT) are the first and rate limiting enzymes for the oxidative and reductive PPP, respectively. Given the importance of de novo nucleotide synthesis in cell proliferation, we evaluated the effects of a G6PD inhibitor (G6PDi1) and a TKT inhibitor (Oxythiamine) on the proliferation of CLSA25 and naïve PDAC cells. As shown in Fig. S4A, B, G6PDi1 abrogated CLSA-mediated cell proliferation in MiaPaCa-2 and PANC1 cells, while Oxythamine had no effect. CLSA PDAC cells were also more sensitive to G6PDi1-induced cell death (Fig. S4C). To determine whether OxPPP might be upregulated by CLSA, we evaluated the protein abundance of G6PD, PGLS, and PGD in CLSA-25 PDAC cells and CLSA-I PDOs. As shown in Fig. 4A, B, G6PD protein levels were increased in all the CLSA-25 cells and CLSA-I PDOs when compared to their naïve controls. The protein levels of PGLS and PGD were also upregulated in CLSA cells and PDOs except for MiaPaCa-2 (for PGLS) and Capan-2 (for PGD) (Fig. 4A, B). Similar upregulation of OxPPP proteins was observed after doxycycline-induced SLC7A11 knockdown in PANC-1 and MiaPaCa-2 cells for 4 weeks (Fig. S4D). Moreover, the upregulation of OxPPP proteins in CLSA-25 MiaPaCa-2 cells did not depend on the continuous presence of CLS (Fig. S4E).

To determine whether CLS in the PDAC patient tumor microenvironment might also lead to PPP upregulation, we stained G6PD and PGD in PDAC tissues and paired para-tumor normal tissue from 18

patients. As shown in Fig. 4C, D, the protein levels of G6PD and PGD were robustly upregulated in PDAC tissues when compared to paired para-tumor pancreatic tissues. To investigate whether OxPPP upregulation might contribute to PDAC progression, we evaluated the abundance of G6PD in PDAC tissues from a cohort of 100 patients. As shown in Table S1, G6PD high patients had larger tumor size, more advanced TNM stages, and higher incidence of lymph node metastasis (Table S1). G6PD high patients also had lower overall survival (OS) and relapse free survival (RFS) (Fig. S4F), which indicates a critical role for OxPPP in PDAC tumor growth and progression.

Next, we investigated the mechanism underlying CLSA-mediated OxPPP upregulation in PDAC. We found no significant changes in the mRNA transcript levels of the three OxPPP genes (G6PD, PGLS, and PGD) in CLSA-25 cells (Fig. S4G). G6PD is a very stable protein, with no detectable degradation even after 12–24 h cycloheximide (CHX) treatment (Fig. S4H). Therefore, we hypothesized that OxPPP enzymes might be translationally regulated in CLSA PDAC cells. We used L-Azidohomoalanine (AHA) to label newly translated proteins in CLSA-25 and naïve MiaPaCa-2 and PANC-1 cells[40]. The labeled peptides were tagged using a biotin-alkyne click-chemistry probe. After pull-down with streptavidin, the abundance of AHA-labeled G6PD, PGLS, and PGD was determined by Western blotting. The levels of newly translated G6PD and PGD were two to fivefold higher in CLSA MiaPaCa-2 and PANC-1 cells when compared to control (Fig. 4E). AHA-tagged PGLS only increased in CLSA-25 PANC-1 cells, but not in MiaPaCa-2 cells (Fig. 4E).

To further investigate the translational regulation of PPP enzymes, a sucrose gradient polysome profiling experiment was carried out

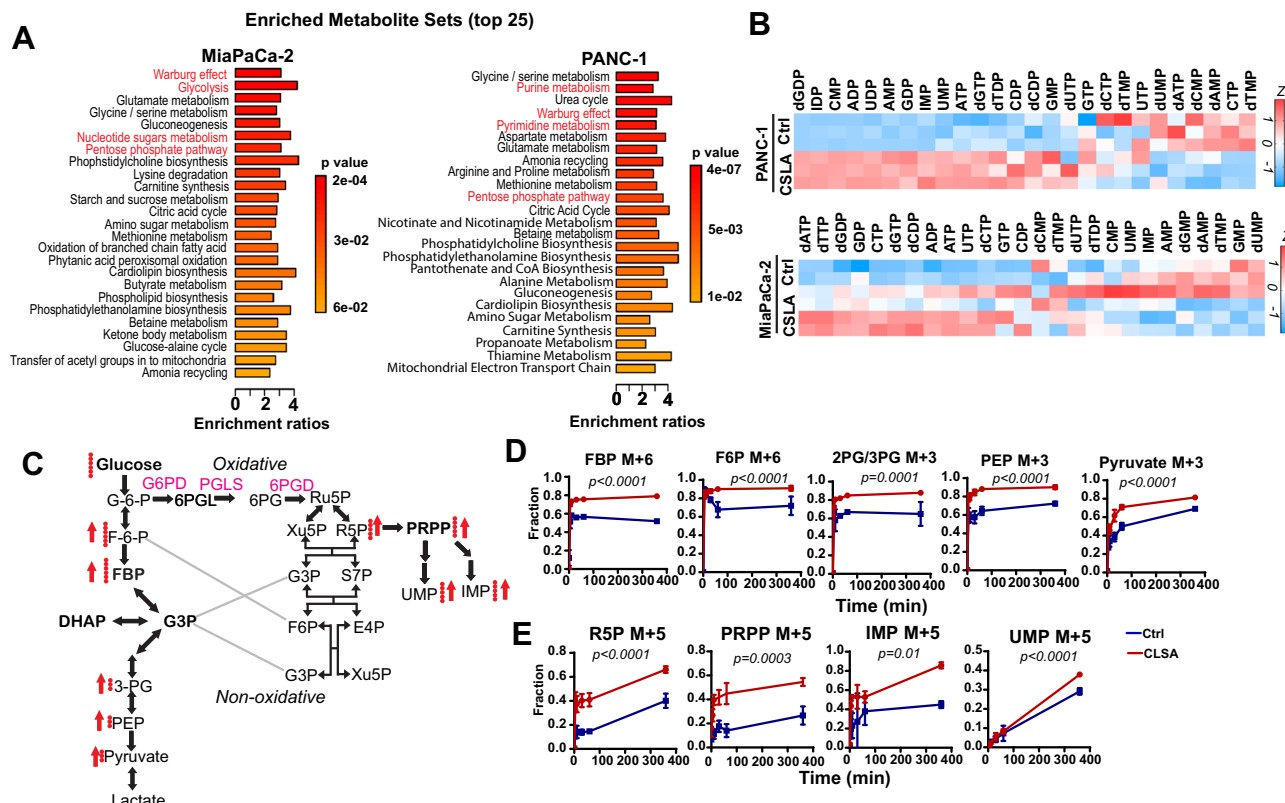

**Fig. 3 | Metabolic reprogramming in CLSA cells activates PPP metabolism and nucleotide synthesis. A** Metabolic Set Enrichment Analysis showing the 25 most differentially regulated metabolic pathways in CLSA-25 MiaPaCa-2 and PANC-1 cells according to targeted metabolomics screening. *p* value was derived from Over-representation analysis through MetaboAnalyst 5.0. **B** Heatmap showing the levels of different nucleotides in CLSA-25 PANC-1 and MiaPaCa-2 cells in metabolomics screening. **C** Summary of $^{13}$C incorporation into glycolysis and PPP metabolites in control and CLSA-25 MiaPaCa-2 cells using [U-$^{13}$C] glucose as tracer. Red arrows indicate increases in $^{13}$C incorporation into the indicated metabolite. **D, E** Fractional labeling of intermediate metabolites in glycolysis (**D**), PPP and nucleotide synthesis pathways (**E**) after tracing with [U-$^{13}$C] for indicated time. Data in (**D**) and (**E**) were analyzed using two way ANOVA. Data are shown as mean ± SD of 5 biological repeats.

(Fig. 4F, G). Polysome profiles showed an overall shift of mRNA into heavier polysomes in CLSA-25 MiaPaCa-2 cells (Fig. 4F), which indicated upregulation of translational initiation. The peak distribution of G6PD and PGD mRNA shifted from fraction 3 in control cells to fraction 6 and 4, respectively, in CLSA-25 cells, which indicated that the number of ribosomes associated with those mRNAs was higher in CLSA-25 cells compared with control cells. In contrast, the peak distribution of PGLS and GAPDH mRNA shifted toward lighter fractions, which is consistent with AHA labeling results in MiaPaCa-2 (Fig. 4G). The shift toward heavier polysomes for the G6PD and PGD mRNAs suggests that OxPPP is translationally upregulated in CLSA cells.

To investigate the role of OxPPP in CLSA-mediated phenotypes, we used RNAi to knockdown G6PD in PDAC cells (Fig. S4I). Both nucleotide levels (Fig. 4H, I) and NADPH/NADP$^+$ratios (Fig. 4J) were increased in CLSA-25 PDAC cells, which was suppressed by siRNA depletion of G6PD (Figs. 4H–J and S4J). G6PD siRNA also inhibited the CLSA-mediated increase in ferroptosis resistance and cell proliferation (Fig. S4K, L). The siRNA results were confirmed with doxycycline-induced knockdown of G6PD (Fig. S4M–O). To determine whether activation of OxPPP is sufficient to mediate CLSA phenotypes, we overexpressed G6PD in naïve PDAC cells (Fig. S4P). G6PD over-expression increased resistance to RSL3- and iFSP1-induced ferroptosis in PDAC cells (Fig. S4Q, R), while cell proliferation was not affected by ectopic G6PD (Fig. S4S). Taken together, our data suggested that the activation of OxPPP is at least partially responsible for CLSA-mediated nucleotide synthesis, cell proliferation, and ferroptosis resistance.

## CLSA PDAC cells are hypersensitive to the lipid metabolism drug lomitapide

To investigate whether CLSA could expose exploitable metabolic vulnerabilities, we performed a drug screening using a metabolism-focus library of more than 800 compounds (Table S2). Seventy-six compounds showed more than 70% inhibition of cell viability in CLSA group (Fig. 5A). Among the most effective drugs were glycolysis/glucose uptake inhibitors (e.g., PFK158, BAY-876) and nucleotide synthesis inhibitors (e.g., Leflunomide, Pemetrexed acid) (Figs. 5A and S5A), which are consistent with upregulation of glycolysis and nucleotide synthesis in CLSA cells. Approximately 30% of the positive hits were compounds targeting lipid metabolism, including many FDA-approved drugs for cardiovascular diseases (e.g., Lomitapide, Mevas-tatin) and type II diabetes (e.g., Troglitazone, Pioglitazone) (Figs. 5A and S5A). Considering the excellent safety profiles of lipid metabolism drugs, several drugs within this group, along with Deni-fanstat (FASN inhibitor) and A939572 (SCD1 inhibitor), were further investigated in naïve and CLSA PDAC cells (Figs. 5B and S5B). Among these drugs, lomitapide showed excellent efficacies toward CLSA PDAC cells and PDOs with IC$_{50}$ in the high nanomolar to low micro-molar range (Fig. 5B, C). Lomitapide-induced cell death in control and CLSA PDAC cells could be partially rescued by pan-caspase inhibitor zVAD-fmk and RIPK1 inhibitor necrostatin-1, but not by ferrostatin-1 (Fig. S5C). Moreover, lomitapide treatment had no effect on lipid peroxidation in either control or CLSA PDAC cells (Fig. S5D). There-fore, lomitapide-induced cell death in PDAC cells likely involved apoptosis and necroptosis, but not ferroptosis.

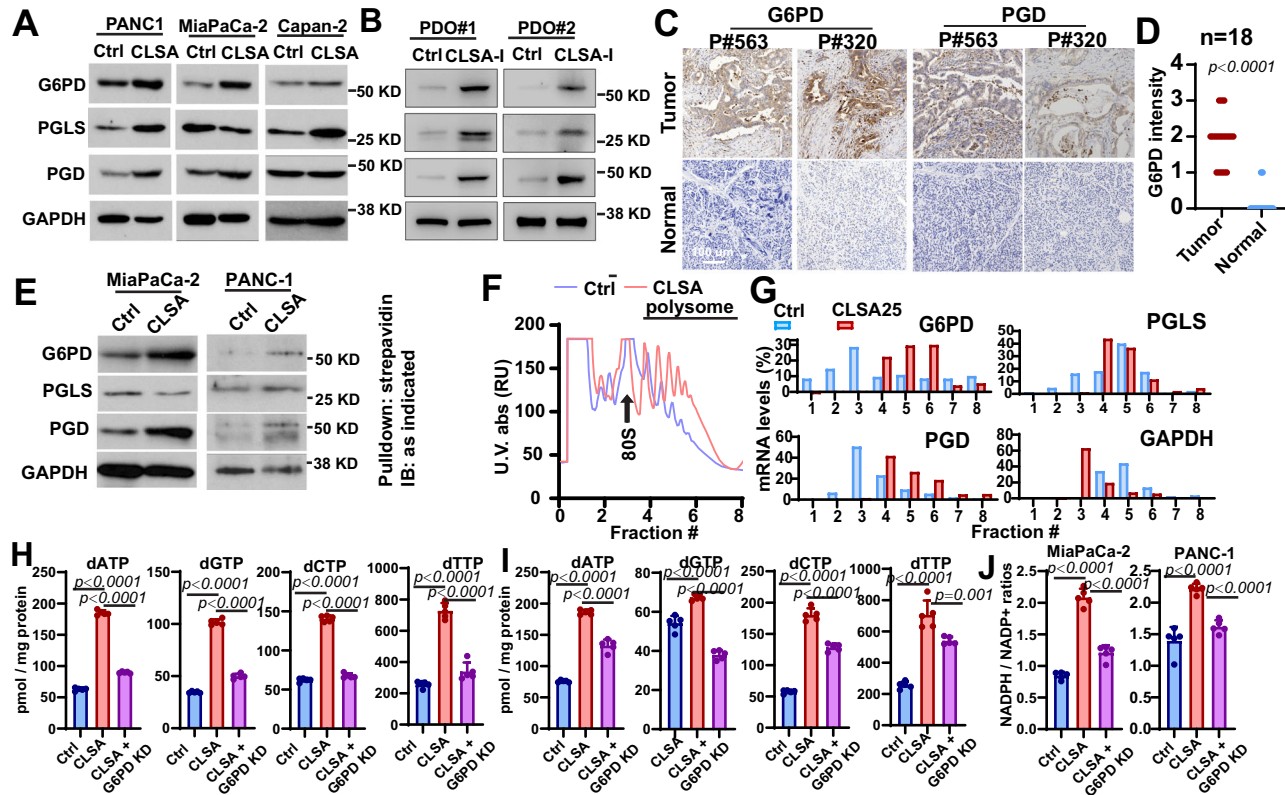

**Fig. 4 | Translational upregulation of OxPPP enzymes promotes nucleotide and NADPH synthesis in CLSA PDAC cells. A, B** Western blotting showing the levels of G6PD, PGLS, and PGD, the three OxPPP enzymes in CLSA-25 PDAC cell lines (**A**) and CLSA-I PDOs (**B**). **C, D** Representative IHC staining (**C**) and quantitation (**D**) showing the levels of G6PD and PGD in PDAC tumor tissues and paired para-tumor normal tissues (*n* = 18). **E** AHA labeling of newly synthesized G6PD, PGD, and GAPDH in Ctrl and CLSA-25 PDAC cells. Cells were pulsed with AHA for 1 h. AHA-incorporated peptides were labeled with Biotin, affinity purified with streptavidin-beads and used for Western blotting analysis. **F** Sucrose gradient polysome profiling analysis showing a shift in polysome profiles in CLSA-25 MiaPaCa-2 cells. **G** qPCR analysis of

the distribution of the mRNA transcripts of G6PD, PGD, PGLS, and GAPDH in fractions collected from CLSA-25 and control MiaPaCa-2 polysome profiling experiments in (**F**). **H, I** LC-MS/MS quantitation of the effects of CLSA-25 and G6PD KD on dNTP levels in MiaPaCa-2 (**H**) and PANC-1 (**I**) cells. **J** LC-MS/MS analysis of the effects of CLSA-25 and G6PD KD on NADPH/NADP⁺ ratios in MiaPaCa-2 and PANC-1 cells. Data in (**D**) were analyzed using two-sample, two-tailed Mann−Whitney *U*-test. Data in (**H**−**J**) were analyzed using two-sided one-way ANOVA followed by Dunnett's multiple comparison test. Data in (**A**, **B**) and (**E**−**G**) are representative results from at least 3 independent experiments. Data in (**H**−**J**) showed mean ± SD of 4 biological repeats.

---

Lomitapide is an orphan drug approved by the FDA for the treatment of homozygous familial hypercholesterolemia (HoFH)[41]. Developed as an inhibitor for microsomal triacylglycerides transfer protein (MTTP), lomitapide inhibits the transfer of TG to ApoB in hepatocytes and enterocytes to suppress the secretion of ApoB-containing lipoprotein into the bloodstream[41,42]. As shown in Fig. 5D−F, orthotopically implanted CLSA-25 MiaPaCa-2 xenograft grew faster than naïve control tumors. The CLSA-25 group was more sensitive to the anti-tumor effect of lomitapide and treatment with 10 mg/kg lomitapide completely abrogated the CLSA effects on promoting tumor growth (Fig. 5D−F).

**Lomitapide induces lipotoxicity by suppressing triacylglycerides synthesis and LD formation**

In an effort to determine the mechanism underlying lomitapide cytotoxicity, we found no detectable protein levels of MTTP in naïve or CLSA PDAC cells (Fig. S5E). Therefore, we performed targeted lipidomics screening to investigate the effect of lomitapide treatment on lipid metabolism in MiaPaCa-2 cells. As shown in Figs. 5G and S5F−J, lomitapide treatment (1 μM) significantly decreased the levels of TG and cholesterol ester (CE) while increasing the levels of free fatty acids (FFA) and acyl-carnitines (CAR), two toxic lipid species. The levels of phospholipids (PL) and lyso-phospholipids (LPL) were also increased after lomitapide treatment (Fig. S5G, H). TLC analysis (Figs. 5H, I and S6A, B) showed that lomitapide treatment decreased TG levels in

PDAC cells in a dose-dependent manner, while the levels of DG were not affected. LD staining and confocal imaging indicated that lomitapide robustly inhibited LD formation in PDAC cells (Fig. 5J, K). It is interesting to note that when compared to DGAT1 inhibitor (DGAT1i) A922500 and DGAT2 inhibitor (DGAT2i) PF-06424439, lomitapide was able to achieve larger reduction of TG levels at much lower concentrations (Fig. S6A, B). Immunofluorescence staining of MiaPaCa-2 xenograft tumor sections showed CLSA increased Perilipin 2 (PLIN2, LD marker) staining by approximately twofold (Fig. S6C). Lomitapide treatment decreased PLIN2 staining intensity in both control and CLSA tumors by approximately twofold (Fig. S6D). Our data indicated that lomitapide treatment reduced LD levels in cell culture and xenograft tumor models. Given recent reports implicating LD in ferroptosis resistance[43,44], we tested the effects of lomitapide on PDAC cell sensitivities to RSL-3 induced ferroptosis. As shown in Fig. S6E, lomitapide (0.5 μM) decreased RSL-3 IC₅₀ by two to fourfold and 15 to 40-fold in naïve and CLSA PDAC cells, respectively. Therefore, although lomitapide doesn't induce ferroptosis as a single agent, it could potentially be used to sensitize PDAC cells to ferroptosis.

To understand the mechanism by which lomitapide regulates TG, we used oleic acid-alkyne (OA-alkyne), a click-chemistry fatty acid probe, to evaluate the effects of lomitapide treatment on de novo TG synthesis. MiaPaCa2 and PANC-1 cells were pre-treated with various concentrations of lomitapide for 2 h and then pulsed with OA-alkyne. After 30 min labeling, the lipids were extracted, labeled with 3-azido-7-

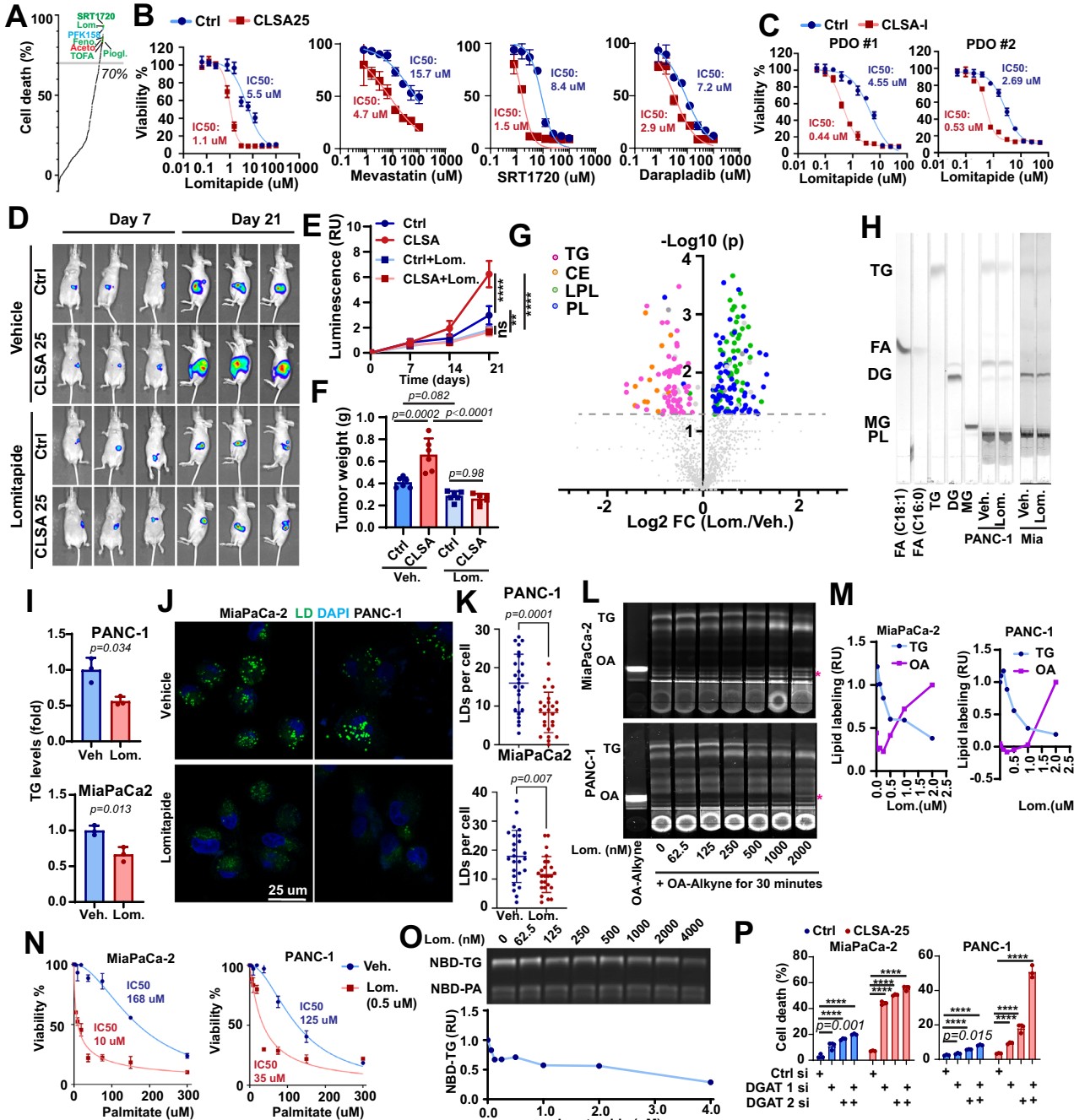

**Fig. 5 | Lipid metabolism in CLSA cells is a metabolic vulnerability targetable by lomitapide. A** Summary of drug screening results. Drugs with reported roles in regulating lipid metabolism, nucleotide metabolism, or glycolysis were labeled was green, red, and cyan, respectively. The grey line indicates 70% of cell death in the high throughput screening. **B** Dose response curves showing the effects of lipid metabolism drugs on the viability of control and CLSA-25 MiaPaCa-2 cells. **C** Dose response curves showing the effects of lomitapide on the viability of control and CLSA-I PDO#1 and 2. **D, E** Representative BLI imaging (**D**) and quantitation (**E**) ($n = 6$, nude mice, mean ± SD) showing the effects of lomitapide treatment (10 mg/kg, oral gavage daily) on the orthotopic xenograft tumor growth of control or CLSA-25 MiaPaCa-2. ns, $p = 0.98$; **, $p = 0.02$; ****, $p < 0.0001$. **F** The weight of xenograft tumors harvested from mice in (**D** and **E**) ($n = 6$, mean ± SD). **G** Volcano plot showing the effects of lomitapide treatment (1 μM, 24 h) on lipidomics in CLSA-25 MiaPaCa-2 cells, as determined through targeted lipidomics screening. TG, CE, PL (phospholipids), and LPL (lysophospholipids) are colored magenta, orange, green, and blue, respectively. **H, I** TLC analysis (**H**) and densitometry quantitation (**I**) to show the effects of lomitapide treatment (4 μM, 24 h) on TG levels in MiaPaCa-2 and PANC-1 cells. **J, K** Confocal microscopy imaging (**J**) and quantitation (**K**) of LD staining to show the effects of lomitapide treatment (2 μM) on LD levels in PANC-1 and MiaPaCa-2 cells. **L, M** TLC analysis (**L**) and densitometry quantitation (**M**) showing the effects of lomitapide treatment on the de novo TG synthesis using OA-alkyne probe as tracer. **N** The effect of lomitapide treatment (0.5 μM) on palmitate-induced lipotoxicity in CLSA-25 MiaPaCa-2 and PANC-1 cells. Cells were treated with palmitate at indicated concentrations with or without lomitapide for 48 h. **O** The inhibition of diacylglycerol O-acyltransferase activity in isolated liver microsomal membrane (5 μg protein) by lomitapide at different concentrations. The lower panel is the densitometry quantitation of TLC analysis in the upper panel. **P** The effects of DGAT1 and DGAT2 knockdown on cell viability in control and CLSA-25 PDAC cells. Data were analyzed using two-sided one-way ANOVA followed by Dunnett's multiple comparison test. Data in (**E**−**G**, **I**, and **K**) were analyzed using two-sample, two-tailed unpaired Student's *t*-test and not adjusted for multiple comparison. Data in (**B**, **C**, **I**, **N**, and **P**) show mean ± SD from 3 biological repeats. Data in (**K**) showed mean ± SD of LD numbers from 25 cells in each group. **** represents $p < 0.0001$. Numerical $p$ values are shown in (**J**) and (**K**) when $p \geq 0.0001$.

hydroxycoumarin through click-chemistry reaction, and the OA-alkyne labeled lipids were resolved through TLC. As shown in Fig. 5L, M, lomitapide inhibited de novo TG synthesis with an $IC_{50}$ of approximately 0.5 μM. Accompanying the dose-dependent decrease in OA-labeled TG, the free, unincorporated OA levels were increased in lomitapide treated PDAC cells (Fig. 5L, M). When [U-$^{13}$C]-OA was used as tracer, lomitapide inhibited $^{13}$C-OA incorporation into neutral lipids, while the labeling of free OA or OA in polar lipids are not affected (Fig. S6F). TG synthesis is essential for cells to neutralize FFA-induced lipotoxicity[22]. Therefore, we determined the effects of lomitapide treatment on sensitivity to palmitate-induced lipotoxicity in PDAC cells. As shown in Fig. 5N, co-treatment with 0.5 μM lomitapide reduced palmitate $IC_{50}$ in CLSA MiaPaCa-2 and CLSA PANC-1 cells from 168 μM and 125 μM to 10 μM to 35 μM, respectively, which suggests that lomitapide treatment increases lipotoxicity sensitivity in PDAC cells.

The inhibition of TG synthesis and accumulation of FFA in lomitapide-treated PDAC cells is reminiscent of similar phenotypes in DGAT1 and DGAT2 knockout mice[45]. Therefore, we investigated the effects of lomitapide on DGAT activities following a previously reported protocol[46]. As shown in Fig. S6G, microsomal membranes isolated from mouse liver were able to synthesize TG using dioleyl glycerol (DOG) and NBD-palmitoyl-CoA as substrate. Using this assay, we determined that lomitapide could inhibit DGAT activation with an apparent $IC_{50}$ of approximately 1 μM (Fig. 5O). It has been previously reported that lomitapide could be processed and degraded by cytochrome P450[41], which is enriched in the microsomal membrane used for the in vitro DGAT activity assay. Pre-incubation of lomitapide with microsomal membranes (from mouse liver, naïve or CLSA-25 MiaPaCa-2 cells) for 1 h or longer completely abrogated its inhibition of DGAT activities (Fig. S6H). Therefore, the inhibitory effect of lomitapide on DGAT activity in this in vitro assay is likely underestimated.

To determine whether the inhibition of DGAT1 and 2 might be responsible for the cytotoxicity of lomitapide, we used siRNA to deplete DGAT 1 and 2 in MiaPaCa-2 and PANC-1 cells. As shown in Fig. S6I–K, DGAT1 or 2 knockdowns significantly reduced the LD staining in control and CLSA-25 PDAC cells. Moreover, CLSA-25 PDAC cells were more sensitive to DGAT1 or 2 siRNA induced cell death than naïve control cells (Fig. 5P). Taken together, our data suggested lomitapide inhibits TG synthesis and LD formation at least in part by targeting DGAT activities in PDAC cells, and CLSA cells are hypersensitive to lomitapide-induced lipotoxicity, likely due to CLSA-mediated activation of lipogenesis.

## CLSA promotes lipogenesis and lipid droplet formation in PDAC

Since CLSA PDAC cells are hypersensitive to lipid metabolism drugs, we speculated that CLSA might affect lipid metabolism in PDAC. Notably, OxPPP has been implicated in promoting lipogenesis in adipocytes and cancer cells[17,47]. NADPH, along with malonyl-CoA, is required for de novo fatty acid synthesis by fatty acid synthase and suppression of OxPPP inhibits lipogenesis in cancer cells[17,18]. We used targeted lipidomics and untargeted lipidomics screening to examine the effect of CLSA on lipid metabolism changes in PANC-1 and MiaPaCa-2 cells, respectively. (Figs. 6A, B and S7A–D). Interestingly, TG was the most upregulated lipid species in both CLSA-25 lines when compared to their naïve controls (Fig. 6A, B). We confirmed through thin layer chromatography (TLC) that TG was upregulated in CLSA-25 PANC-1 and MiaPaCa-2 cells (Fig. 6C, D). Confocal microscopy indicated that CLSA promotes LD formation (Fig. 6E–G).

Lipase activity assay showed that there was no change in the activities of ATGL, the major lipase in TG lipolysis (Fig. S8A), in CLSA PDAC cells, which suggested that the elevated TG and LD in the adapted cells were not due to inhibition of lipolysis. G6PD overexpression (OE) in naïve MiaPaCa-2 and PANC-1 cells was sufficient to increase TG and LD levels (Fig. S8B–E). Conversely, G6PD knockdown in CLSA PDAC cells significantly decreased TG levels (Fig. 6H, I). These

data suggested that CLSA promotes lipogenesis and LD formation through activation of the OxPPP. To further critically investigate this notion, we used $^{13}$C-glutamine labeling to determine the effects of CLSA and G6PD inhibition (with G6PDi1) on de novo FA synthesis. As shown in Fig. 6J, K, $^{13}$C-labeled palmitate (PA, C16:0) and stearate (SA, C18:0) were increased in CLSA-25 PANC-1 cells when compared to naïve control cells, especially those species with more than 8 $^{13}$C atom incorporations (M + x, x ≥ 8). These results indicated the activation of de novo FA synthesis in CLSA PDAC cells. The inhibition of the OxPPP pathway with G6PDi1 suppressed de novo PA and SA synthesis and largely abrogated the effects of CLSA-25, especially in PA and SA species with greater than 12 $^{13}$C atom incorporations (M + x, x ≥ 12). Taken together, our data strongly indicated that CLSA-25 promotes de novo lipid synthesis in an OxPPP-dependent manner.

Since G6PD overexpression promotes TG synthesis and LD formation, we further determined the effects of OxPPP activation on lomitapide sensitivity. As shown in Fig. S8F, G6PD overexpression reduced the lomitapide $IC_{50}$ by approximately twofold, which indicated that activation of OxPPP was sufficient to sensitize PDAC cells to lomitapide.

To determine whether LD formation is upregulated in PDAC, we performed LD staining in PDAC tissue cryosections from 30 patients. LD staining in PDAC patient tissues and paired para-tumor normal tissues revealed dramatic increase in LD formation in tumor tissues (Figs. 6L, M and S8G). PLIN2 was also upregulated in PDAC tissues when compared to paratumor normal (Fig. S8H). In PDAC tissues, LD almost exclusively colocalized with CK19 positive PDAC cells, but not with α-SMA positive cancer-associated fibroblast cells (Fig. S8G). LD levels in PDAC patient tissues significantly correlated with G6PD staining intensities (Fig. S8I, J). Taken together, our data indicated that OxPPP upregulation in CLSA PDAC promotes lipogenesis and LD formation.

## Lomitapide inhibits chemotherapy-induced LD-formation and TG synthesis and sensitizes PDAC tumors to modified FOLFIRINOX in PDX and KPC models

TG synthesis and LD formation are critical for cells to avoid stress-induced lipotoxicity[21,23]. Targeted lipidomics analysis of MiaPaCa-2 cells treated with modified FOLFIRINOX (mFOLFIRINOX)[48] chemotherapy showed that mFOLFIRINOX treatment induced dramatic lipidome reprogramming that increased TG levels (especially the levels of TG containing long chain FA and PUFAs) while reducing the levels of PL and LPL (Figs. 7A, B and S9A–E). This observation is consistent with the notion that chemotoxic stress induces lipolysis and stores excess FFA in TG and LD to mitigate lipotoxicity[21,23]. We confirmed that mFOLFIRINOX treatment increased TG and LD levels (Figs. 7C and S10A, B). mFOLFIRINOX-induced TG synthesis and LD formation was abrogated by lomitapide treatment (Figs. 7C and S10A, B). Co-treatment of PDOs with mFOLFIRINOX and 200 nM lomitapide synergistically reduced mFOLFIRINOX $IC_{50}$ by approximately two to fivefold in control or CLSA-I PDOs, respectively (Figs. 7D and S10C). Chemo-sensitization effects of lomitapide could be observed at a concentration as low as 50 nM (Figs. 7D and S10C). Similar chemo-sensitization effects were observed when lomitapide was combined with the Abraxane/Gemcitabine chemotherapy regimen (Fig. S10D). The knockdown of DGAT1 or DGAT2 also dramatically increased chemo-sensitivity (Fig. S10E). Conversely, inhibition of TG lysis with ATGL inhibitor decreased chemosensitivity (Fig. S10F, G). Taken together, our data suggested that lomitapide sensitizes PDAC to chemotherapy by targeting DGAT-mediated TG synthesis and LD formation.

To determine whether lomitapide could be used to improve the efficacy of FOLFIRINOX therapy, we used PDX models derived from two PDAC patients (Fig. 7E). NSG mice bearing PDX tumor fragments were administered with vehicle control, lomitapide, mFOLFIRINOX or

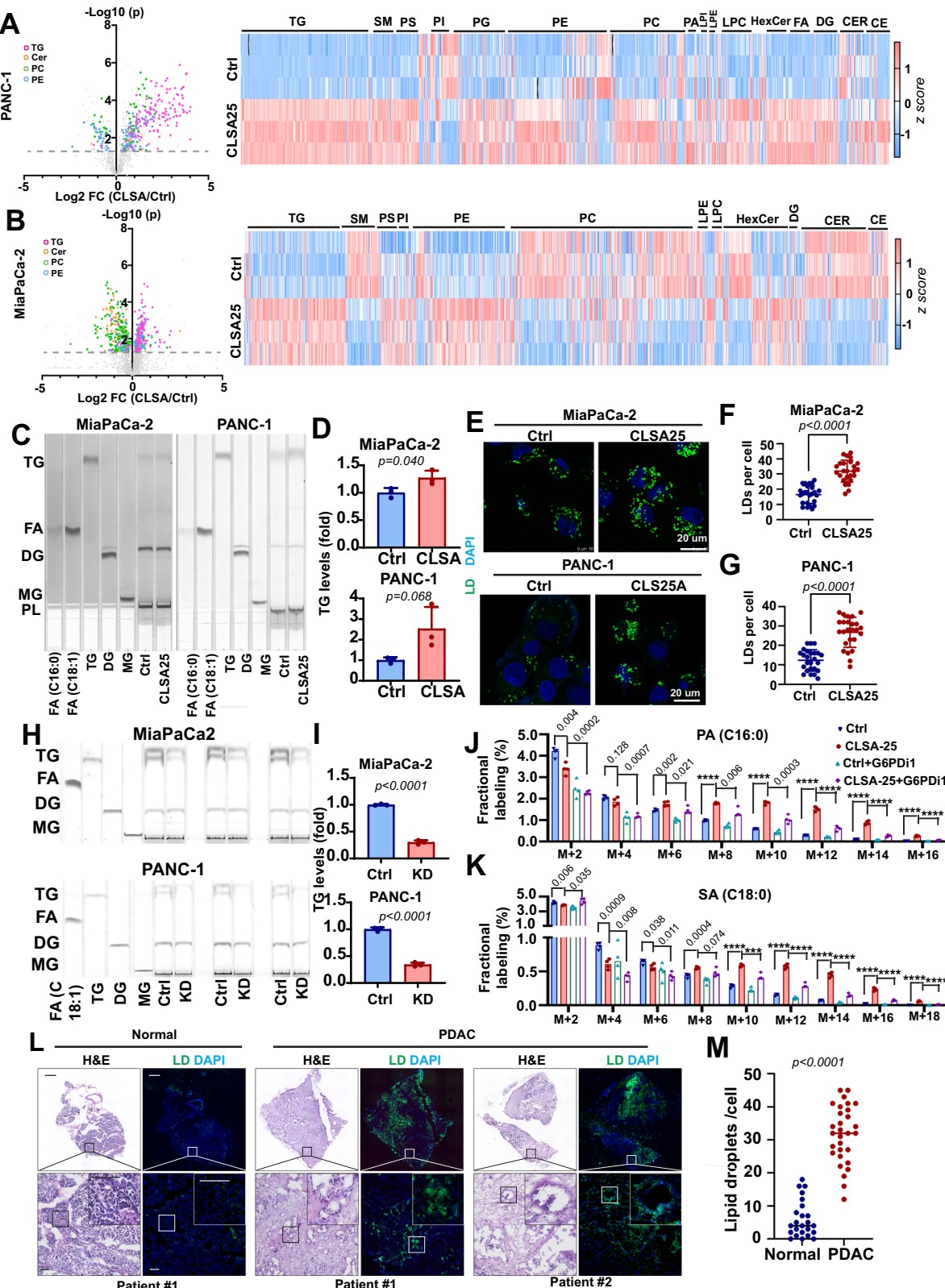

the combination treatment when the tumor size reached 3 mm in diameter (Fig. 7E). As shown in Fig. 7F, G, mFOLFIRINOX treatment inhibited the tumor growth in PDX-1 and PDX-2 mice by 52% and 16%, respectively. Lomitapide treatment alone inhibited PDX-1 and PDX-2 growth by 40% and 35%, respectively. When combined with mFOL-FIRINOX, the inhibition of tumor growth was further enhanced to 78% and 77% (Figs. 7F, G and S10H). The inhibitory effects of single and

combination treatments on tumor growth were confirmed with positron emission tomography scan using $^{18}$F-fluorodeoxyglucose as tracer (Figs. 7H, I and S10I).

To further evaluate the chemo-sensitization effects of lomitapide, we tested the effects of lomitapide and mFOLFIRINOX treatments in the genetically engineered KPC (*LSL-Kras^{G12D/+}; LSL-TrpS3^{R172H/+}; Pdx1-Cre*) PDAC model. The PDAC tumor formation in KPC mice was

**Fig. 6 | CLSA promotes de novo lipogenesis, TG synthesis, and LD formation by upregulating OxPPP. A, B** Volcano plot and heatmap showing that CLSA-mediated lipidomics reprogramming upregulates TG levels in PANC-1 (**A**) MiaPaCa-2 (**B**) cells. **C, D** TLC analysis (**C**) and densitometry quantitation (**D**) showing the effects of CLSA-25 on TG levels in MiaPaCa-2 and PANC-1 cells. **E−G** Confocal microscopy imaging (**E**) and quantitation (**F** and **G**) of LD staining showing upregulation of LD formation in CLSA-25 PDAC cells. Data in (**F** and **G**) are shown as mean ± SD of LD numbers in 25 cells each. **H, I** TLC analysis (**H**) and densitometry quantitation (**I**) showing the effects of G6PD knockdown (KD) on TG levels in CLSA-25 MiaPaCa-2 and PANC-1 cells. **J, K** The effect of CLSA-25 and G6PD inhibition (G6PDi 50 μM) on de novo synthesis of palmitic acid (PA) (**J**) and stearic acid (SA) (**K**), as determined through [$^{15}N_2$, $^{13}C_5$]-Glutamine tracing (24 h) followed by LC-HRMS analysis of fractional $^{13}C$ incorporation. **L, M** Representative confocal micrograph (**L**) and quantitation (**M**) showing BODIPY 493/503 LD staining levels of PDAC tissue cryosections and paired para-tumor normal pancreas tissues ($n = 30$). The scale bars in (**L**) are 1 mm (top) and 100 μm (middle and bottom). Data in (**A** and **B**) were analyzed using two-sided, two-sample $t$-test and not adjusted for multiple comparison. Data in (**F**, **G**, and **M**) were analyzed using two-sample, two-sided Mann−Whitney $U$-test. Data in (**D**, **I−K**) showed mean ± SD of 3 biological repeats and were analyzed using two-sample, two-sided $t$-test. **** represents $p < 0.0001$, respectively. Numerical $p$ values are shown in (**J**) and (**K**) when $p \geq 0.0001$.

monitored by ultrasound imaging. Mice were randomly assigned into one of the four treatment groups (vehicle, lomitapide, mFOLFIRINOX, or combination) after confirmation of tumorigenesis. The mice were treated for 4 weeks, and the tumor growth was monitored through ultrasound during this period. After stopping treatment at week 4, the mice were further monitored for survival. As shown in Fig. 7J–L, after 4 weeks of treatments, lomitapide and mFOLFIRINOX reduced KPC tumor volume by 23% and 43%, respectively. The combination treatment reduced tumor size by 78% (Fig. 7J, K). As a monotherapy, lomitapide improved the median survival from 52 days (in control group) to 70 days ($p = 0.04$). The median survival in the mFOLFIRINOX group was 65 days ($p = 0.20$). The combination treatment further increased the median survival to 121 days ($p = 0.001$) (Fig. 7L). Therefore, lomitapide could inhibit tumor growth and improve animal survival in KPC model as a single agent. When combined with mFOLFIRINOX, lomitapide robustly increased the chemotherapy efficacy and prolonged animal survival. Importantly, although mFOLFIRINOX treatment modestly reduced the body weight of KPC mice, lomitapide treatment, alone or in combination with mFOLFIRINOX, had no additional effects on weight loss (Fig. S10J). These data indicated that lomitapide and its combination with mFOLFIRINOX were well tolerated in mice.

PLIN2 staining of tumor tissues collected from KPC mice showed that mFOLFIRINOX increased LD levels, which could be inhibited by lomitapide (Fig. 7M, N). LD staining in PDX tissue cryosections showed that mFOLFIRINOX treatment significantly increased LD levels, which could be inhibited by lomitapide (Fig. S10K–L).

Taken together, our data suggested that lomitapide has potent chemo-sensitization effects, potentially through inhibiting TG synthesis, LD formation, and enhancing chemotherapy-induced lipotoxicity.

## Discussion

Cys and Cys$_2$ are among the most depleted amino acids in the PDAC tissues and there is intense interest in developing Cys$_2$ deprivation agents as potential anti-cancer therapies. However, the effects of chronic CLS adaptation on cancer cells and the underlying mechanisms are poorly understood. Our data uncovered CLS-induced metabolic reprogramming in PDAC cells, including downregulation of mitochondrial OXPHOS, increased glycolysis, pentose phosphate pathway metabolism, nucleotide synthesis, and de novo lipogenesis. As a result of this metabolic reprogramming, CLSA promotes PDAC cell proliferation, tumor growth, and metastasis.

The metabolic reprogramming in CLSA PDAC was in part due to suppression of Fe-S synthesis and translational activation of the OxPPP. While genetic or pharmacological inhibition of G6PD abrogated the CLSA-mediated tumor cell proliferation and ferroptosis resistance, ectopic expression of G6PD could only partially recapitulate some of the CLSA phenotypes (e.g., ferroptosis resistance, lipogenesis), but not others (e.g., cell proliferation). These findings implicated the involvement of other metabolic pathways in addition to OxPPP during CLSA. A common adaptation response to chronic amino acid deprivation stress is the compensatory activation of the mTORC1, which promotes both nucleotide and lipid synthesis. The potential role of mTORC1 in CLSA will need to be further investigated in the future.

Through unbiased drug screening, we identified lipid metabolism as a metabolic vulnerability in CLSA PDAC. Among FDA-approved lipid metabolism drugs, lomitapide was selected for further investigation. Instead of targeting MTTP, our data showed that lomitapide inhibits TG synthesis and LD formation to sensitize PDAC cells to lipotoxicity. Lomitapide has been previously reported to inhibit mTORC1 and ZDHHC5[49,50]. However, the lomitapide concentrations required for such inhibitions (25 μM) are at least one order of magnitude higher than required for the induction of cytotoxicity in PDAC cells and PDOs. In contrast, the IC$_{50}$ of lomitapide toward TG synthesis (approximately 500 nM in OA-alkyne tracing experiment) was comparable to concentrations required for its cytotoxic effects in PDAC cell lines and tumoroids.

Targeting the dysregulated cancer metabolism is an area under intensive investigation. Drugs targeting nucleotide synthesis, such as gemcitabine and 5-FU have been successfully used in cancer treatment for decades. However, the development of cancer therapeutics targeting dysregulated lipid metabolism was more challenging. Recently, the role of DGAT1/2 and LD emerged as critical players in tumor growth and progression. Several pharmacological inhibitors for DGAT1/2 have been used in clinical trial for diabetes, fatty liver disease, and other metabolic conditions[51,52]. These inhibitors could potentially be useful in targeting the hyperactive lipid metabolism in cancer. Indeed, there is evidence supporting the anti-tumor activity of DGAT1 inhibitor A922500 in glioblastoma and clear cell renal cell carcinoma models[20,53]. However, the doses of DGAT1 inhibitor required for anti-tumor activities in mouse models were 20- to 40-times higher than those typically used for other metabolic diseases (3 mg/kg)[52]. The feasibility of using such high doses in cancer patients is likely an issue.

Importantly, lomitapide robustly inhibited tumor growth and improved the efficacy of FOLFIRINOX chemotherapy regimen when used at a clinically relevant dose. Even in naïve PDAC cells and organoids, lomitapide still has significant lipotoxic and chemo-sensitizing effects, which suggested that it has wider applicability beyond CLSA PDAC. The lomitapide dose used in our pre-clinical animal experiments (10 mg/kg) is equivalent to approximately 0.8 mg/kg in humans, which has been previously shown to be safe for HoFH patients for at least 78 weeks in clinical trials[41]. Considering that a typical FOLFIRINOX chemotherapy session lasts only 2 weeks, it is possible that higher doses of lomitapide could be tolerable as chemo-sensitizing agents. Therefore, lomitapide could serve as a powerful agent for targeting the dysregulated lipid metabolism in PDAC and other cancers.

## Limitations of the current study

While our data suggested a CLSA program in PDAC that promotes PDAC tumor growth and metastasis, these findings are mostly based on in vitro cell or organoid culture models. In the PDAC micro-environment, tumor cells are likely subjected to a combination of stress factors, including deprivation of other nutrients, hypoxia, acidosis, etc. Furthermore, due to the heterogenous nature of the PDAC TME, the CLSA response in PDAC could be complicated and

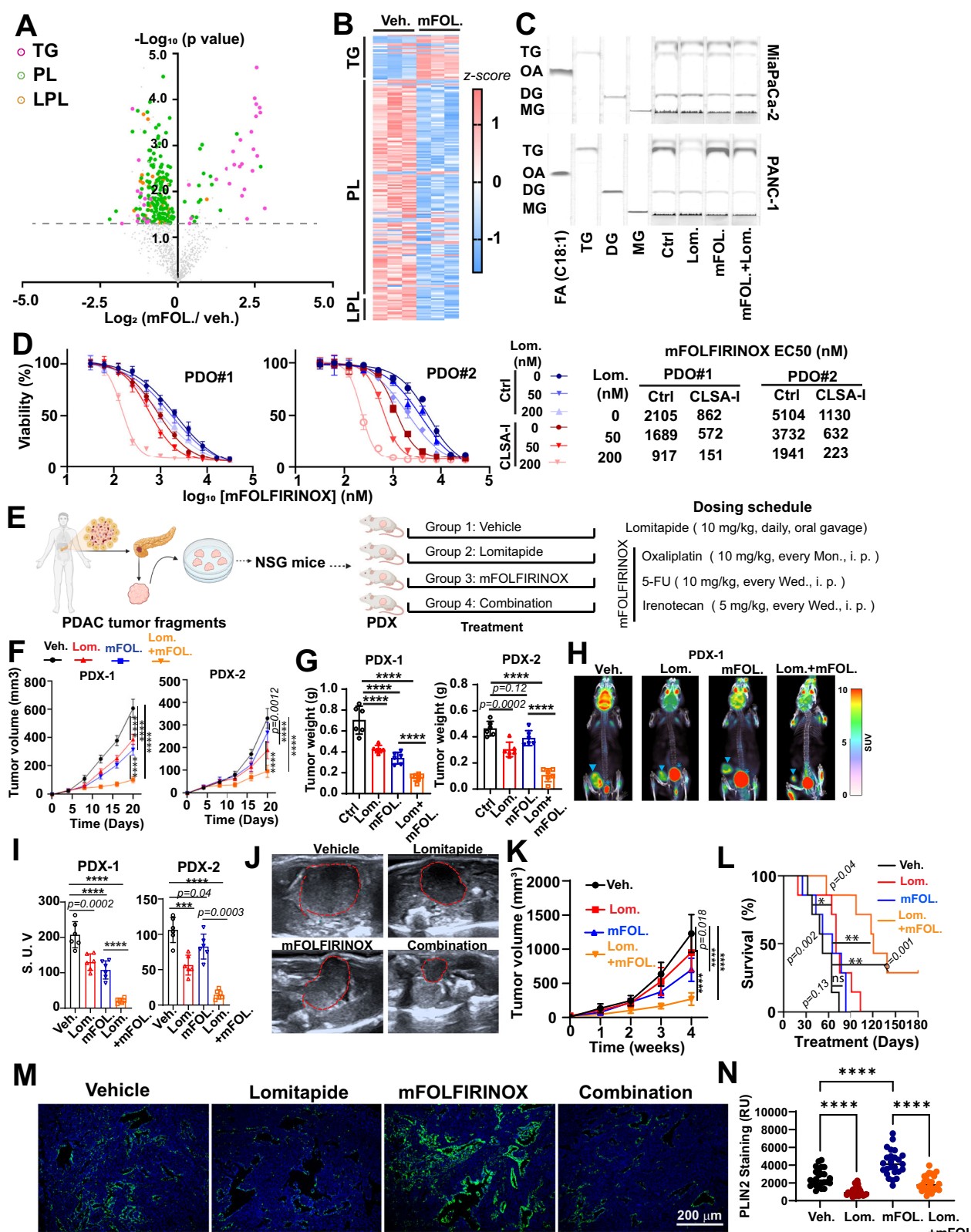

context dependent. Future investigation into the CLSA response in vivo would be critical to understand the role of CLS in PDAC progression.

There are many remaining questions that need further investigation. First, the cause(s) of CLS in PDAC remained incompletely understood. MS imaging indicated progressive downregulation of Cys from normal tissues, PanIN to PDAC in the KPC model. This could be

due to the increased demand, higher redox flux, and limited supplies (due to desmoplasia) in PDAC. Interestingly, tumor infiltrating lymphocytes in PDAC tissues had high levels of Cys. It remained unclear whether infiltrating lymphocytes in PDAC, such as Treg, TAM, and/or MDSC might contribute to CLS by competing with tumor cells of Cyst(e)ine. Second, our data showed that CLSA phenotypes, once established, no longer depend on the continuous presence of external

**Fig. 7 | Lomitapide inhibits chemotoxic stress-induced LD formation and enhances the efficacy of FOLFIRINOX regimen in PDAC. A, B** Volcano plot (**A**) and heatmap (**B**) showing the effects of mFOLFIRINOX (8 μM) treatment for 24 h on the levels of TG, phospholipids (PL), and lysophospholipids (LPL) in MiaPaCa-2 cells, as determined by LC-MS/MS targeted lipidomics screening. mFOLFIRINOX cocktail consists of 5-FU (8 μM), Oxaliplatin (8 μM), and SN-38 (4 μM). Grey line indicates $p = 0.05$. **C** TLC analysis showing the effects of mFOLFIRINOX (8 μM) and lomitapide (4 μM) treatment (24 h) on TG levels in MiaPaCa-2 and PANC-1 cells. **D** The effects of low dose lomitapide (50 or 200 nM) treatment on sensitivities to mFOLFIRINOX treatment in PDO1 and PDO2. PDOs were treated with the drug for 48 h. Data are shown as mean ± SD of 3 biological replicates. **E** Schematic illustration (generated using BioRender) of the establishment of PDX mouse models and treatment regimen of these mice with lomitapide, mFOLFIRINOX, or the combination of both. Created in BioRender. Tam, J. (2025) https://BioRender.com/ajyfclu. **F–I** The effects lomitapide, mFOLFIRINOX or the combination treatment on tumor growth (**F**), tumor weight (**G**), and $^{18}$F-DG PET-CT scan (**H** and **I**). **H** shows the representative images of $^{18}$F-DG PET-CT scan from PDX1 group and **I** shows the quantification of standardized uptake value (S.U.V.) from PDX1 and PDX2 groups. Arrowhead in (**H**) indicates PDX xenograft tumor and * indicates bladder ($n = 6$ mice per group). Data are shown as mean ± SD. **J–L** The effects lomitapide, mFOLFIR-INOX, or the combination treatment on the tumor growth and survival in KPC genetically engineered PDAC mouse model. Representative ultrasound scan images 4 weeks after treatment (**J**), quantitation of tumor volume determined through ultrasound imaging (**K**), and survival of the mice (**L**) are shown ($n = 7$ mice per group). **M, N** Representative confocal microscopy images (**M**) and quantitation (**N**) of PLIN2 immunofluorescence staining using tumor tissues harvested from KPC experiments in (**J–L**). Data in (**N**) are shown as mean ± SD of 25 ROIs each. Data in (**A**) were analyzed using two-sided two-sample $t$-test without adjustment for multiple comparison. Data in (**G**, **I**, and **N**) were analyzed using two-sided one-way ANOVA followed by Sidak's multiple comparison test. Data in (**F**) and (**K**) were analyzed using two-way ANOVA followed by Tukey's multiple comparison test. Data in (**L**) were analyzed using two-tailed log-rank test. **** represents $p < 0.0001$. Numerical $p$ values were shown when $p \geq 0.0001$. All error bars are shown as mean ± SD.

CLS pressure. The mechanisms underlying such stable and cell autonomous adaptation program require further investigation. Third, while our study focuses on de novo lipogenesis and TG synthesis, our lipidomics screening indicated that CLSA affects the levels of many other lipid subclasses, as well as the saturation and carbon length of the FA side chains. The effects on other aspects of lipidome could have critical implications for CLSA phenotypes, which we are not able to dissect within the scope of this study due to the immense complexity of the lipidomics changes. Fourth, although lomitapide is an FDA-approved orphan drug for HoFH, it could potentially cause mild to moderate gastrointestinal side effects and fat accumulation in the liver[41,54–56]. These side effects are due to its inhibition of MTTP in hepatocytes and enterocytes[41,54–56]. Therefore, future development of lomitapide-based DGAT inhibitors without MTTP inhibitory activity would be highly desirable.

## Methods

### Antibodies, key reagents, and inhibitors
Information for antibodies, key reagents, and inhibitors are included in Tables S3 and S4, respectively.

### Cell culture
The PDAC cell lines (MiaPaCa-2, PANC-1, SW-1990, Capan-2) were obtained from the ATCC. KPC cells were derived from $Kras^{lsl-Kras\ G12D/+}$; $Trp53^{R172H/+}$; $Pdx1$-$Cre$ (KPC) genetically engineered mouse model. Pan02 cell was a gift from Dr. Shari Pilon-Thomass (Moffitt Cancer Center). All ATCC cell lines were characterized or authenticated by the ATCC using short tandem repeat profiling and passaged in our laboratory for fewer than 6 months before use. All cell lines used were free of microbial (including mycoplasma) contamination. We routinely perform test for mycoplasma contamination in the lab at least monthly. All pancreatic cells were cultured in DMEM medium supplemented with 10% fetal bovine serum and 1% Penicillin-Streptomycin and maintained at 37 °C in a humidified 5% $CO_2$ incubator.

### Induction of CLSA
CLSA-25 and CLSA-50 cell culture media were prepared by supplementing pyruvate, glutamine, methionine, and cystine-free DMEM media (Gibco, 21013-024) with 1 mM Sodium Pyruvate, 4 mM glutamine, 200 μM L-Methionine, and 25 μM L-cystine (for CLSA25) or with 50 μM L-cystine (for CLSA50). To induce CLSA, PDAC cells were cultured in indicated L-cystine limited media supplemented with 10% FBS and 1% Penicillin-Streptomycin for 4 weeks. The media were changed every other day during this period. After 4 weeks adaptation, cells were frozen and stored in liquid nitrogen. After recovery from liquid nitrogen cells were cultured in their respective induction media and used for experiments within 6 passages.

### Patient derived organoid (PDOs) culture
PDO were derived from 2 patients (both male and 65–75 years old) and cultured following a previously described protocol[30]. To induce CLSA-I PDO, two PDOs were cultured in medium containing 0.5 μM IKE for 4 weeks. For PDO xenografts, CLSA-I or control PDOs were digested into single cells using TrypLE Express and then neutralized with Human Complete Feeding Medium contained Human Wash Medium with BSA (Advanced DMEM/F-12, 10 mM HEPES, 1 × GlutaMAX Supplement, 100 μg/ml Primocin, and 0.1% BSA). $1 \times 10^5$ of PDO cells were resuspended into Matrigel and injected orthotopically into pacreas. The growth of tumors were monered through MRI scan using 9.4 T MR scanner (Bruker BioSpec 94/30 USR) at Multimodality Preclinical Molecular Imaging Center in Tianjin Medical University General Hospital. The tumor volume was calculated with the formula: volume = length × width × width/2.

### Treatment of PDAC cells and PDOs with modified FOLFIRINOX (mFOLFIRINOX)
mFOLFIRINOX Treatment of PDAC cells and PDO cultures were carried out as previously described. The mFOLFIRINOX cocktail consisted of 5-FU, SN-38 (metabolic product of Irenocan responsible for its inhibitory activity toward DNA topoisomerase I) and oxaliplatin at a 2:1:2 ratios. The indicated mFOLFIRINOX concentrations were the concentrations of 5-FU and oxaliplatin.

### Cell proliferation assay
$1 \times 10^5$ Cells per well were seeded onto a 12-well plate in triplicate. The media were changed every two days. At indicate time points, the cells were fixed with 10% formalin for 15 min and stained with 0.5 mL 10% crystal violet solution for 20 min with shaking. After staining, the cells were washed with water for five times and dried overnight. The crystal violet staining was solubilized with 1 ml 10% acetic acid absorbance at OD 590 nm was determined with a plate reader.

### Cell viability assay
$1 \times 10^4$ cells per well were seeded onto 96-well plates (Corning, 3610) and treated with different drugs at indicated concentrations for two to three days. Cell viabilities were tested using CellTiter-Glo Cell Viability Assay (Promega, G7572) and recorded as relative luminescence units using FlexStation 3 (Molecular Devices).

### 2-NBDG Glucose uptake assay
PDAC cells were seeded at $2 \times 10^5$ cells/well in a 12 well plate. The cells were incubated with 500 μL of culture medium containing 50 μM 2-NBDG and incubated at 37 °C for 2 h. The reaction was terminated by washing the cells twice with ice-cold PBS. The cell was re-suspended

through trypsinization and analyzed through flow cytometry using the excitation/emission at 467/542 nm.

## Mitochondria stress test and glycolysis stress test

Oxygen consumption rate (OCR) and extracellular acidification rate (ECAR) were determined using Seahorse XFe24 Extracellular Flux Analyzer (Agilent Seahorse Bioscience) following protocols recommended by the manufacturer. Cells were seeded on XFe24 cell culture microplates ($2.5 \times 10^5$ cells/well). Cells were maintained in a XF DMEM assay medium (Agilent, 103575-100) in a non-$CO_2$ incubator for 30 min before the assay. The XF Cell Mito Stress Test Kit (Agilent, 103015-100), was used for the OCR assay. XF Glycolysis Stress Test Kit was used for the ECAR assay (Agilent, 103020-100). For mitochondria stress test, the baseline recording was followed by sequential injection the following reagent to the final concentration: 1.5 μM oligomycin, 1 μM FCCP, and 0.5 μM rotenone/antimycin A. For glycolysis stress test, the baseline recording was followed by sequential injection the following reagents: 10 mM D-glucose, 1 μM oligomycin, and 50 mM 2-DG. At the end of the experiment, the cells in each well were lysed with RIPA buffer and the protein concentrations were determined through BCA assays. The ECAR and OCR data were normalized against total protein amount in each well.

## RNA interference and other constructs

The knockdowns of G6PD, DGAT1, and DGAT2 was achieved using the 27mer duplexes DsiRNA system from IDT. The targeting sequences are listed in Table S4. pLX304-G6PD was constructed by subcloning the human G6PD cDNA into pLX304 vector through Gateway subcloning. The lentiviral particles were packaged in HEK293 cells using the PEI transfection method, and concentrated as previously described[57]. After selection with blasticidine, the surviving cells were pooled together and used for assays. siRNA and shRNA target sequences were included in Table S5.

## Tet-inducible knockdown of SLC7A11 and G6PD

For inducible knockdown of SLC7A11 and G6PD, shRNA sequence targeting the two genes (see Table S4) were cloned into the pLVX-H1-TetOne-Puro vector and lentivirus particles were prepared as described in the previous section. After viral particle infection and selection with puromycin, the expression of shRNA were induced by treating cells with 1–100 ng/ml doxycycline for 4–7 days (for acute knockdown) or 4 weeks (for chronic knockdown of SLC7A11).

## Western blotting

Cells were lysed in SDS-NP40 buffer (50 mM Tris, pH 8.0, 150 mM NaCl, 1% NP40, 1% SDS, 1 mM protease inhibitors cocktail) on ice for 1 min. Cells were scraped from the plate and sonicated briefly three times. Then lysates were centrifuged at $20,000 \times g$, 4 °C for 10 min. 20-40 μg proteins were separated by SDS-PAGE and transferred onto PVDF membrane. The membranes were incubated in blocking buffer (5% (w/v) nonfat dry milk in Tris-buffered saline, 0.05% Tween 20 (TBS-T)) for 30 min at room temperature and incubated with primary antibodies for 20 h at 4 °C, followed by incubation with secondary antibodies for 60 min at room temperature. Antibody information are listed in Supplementary Table S3.

## Reverse transcription-polymerase chain reaction

$1 \times 10^6$ cells were washed with ice-cold PBS and total RNA was extracted using RNeasy kit (Qiagen, Cat#74106). 1 μg RNA was used for the reverse transcription using High-Capacity cDNA Reverse Transcription Kit (Thermo Fisher Scientific, 4368814). cDNA was diluted 1:10 then used for qPCR with ABI qPCR master mix (Thermo, Cat#4367659). Each RT-PCR experiment was performed independently at least three times. The PCR primers used are listed in Supplementary Table S6.

## L-Azidohomoalanine (AHA) labeling of OxPPP enzymes

$7 \times 10^6$ cells on a 10 cm dish were rinsed twice with 5 ml 1× PBS and incubated with L-Met free DMEM medium for 1 h in 37 °C, 5% $CO_2$ incubator. After 1 h, L-AHA were added to the final concentration of 1 mM. After 1 h labeling with AHA, cells were lysed. To label newly-synthesized proteins, lysate was mixed with 2× click-chemistry reaction buffer (2 mM Copper Sulfate, 20 μM Biotin-Alkyne, 2 mM TCEP in 1× PBS) and incubated for 60 min at room temperature with shaking. The reaction was terminated by adding TCA to the final concentration of 10% and incubated on ice for 30 min. After $10,000 \times g$ centrifugation at 4 °C 10 min, the precipitate was washed with 1 ml ice-cold acetone for 3 times, and air-dried at room temperature for 30 min. The pellet was re-dissolved in 500 μl RIPA buffer with proteinase inhibitor cocktail and sonicated for 10 s on ice 3 times. The biotin-labeled proteins were precipitated with 40 μl NeutrAvidin resin and affinity-purified OxPPP enzymes were detected with Western blotting using antibodies for G6PD, PGD, PGLS, and with GAPDH as control.

## Polysome profiling

$1 \times 10^7$ cells were treated with CHX (100 μg/ml) in the cell culture media for 2 min and the cells were placed on ice. After removal of media, the cells were washed twice with ice-cold 1× PBS/CHX (100 μg/ml). Cells were scraped in 600 μl Polysome Buffer (50 mM HEPES, pH 7.4, 250 mM KCl, 5 mM $MgCl_2$, 250 mM sucrose) containing 1% Triton ×-100, 1.3% NaDOC, 100 μg/ml CHX, 0.1 U/μl SUPERase. In RNAase inhibitor (AM2694, Invitrogen). After 10 min lysis at 4 °C with rocking, the lysates were centrifuged at $3000 \times g$ for 15 min at 4 °C. 600 μL supernatant was loaded to a sucrose gradient (20%–47%, w/w) and centrifuged in a Beckman SW41Ti rotor at 34,000 rpm for 160 min at 4 °C. After centrifugration, 8 fractions were collected from each. The RNA from each fraction was extracted using TRIzol (15596018, Invitrogen) and used for qPCR analysis.

## LC-MS/MS analysis of NEM-cysteine, GSH, dNTP, and NADP+/NADPH

**NEM-cysteine.** The measurement of NEM-derivatized cysteine in pancreatic cancer cells was carried out following our previously reported protocol[30]. Human samples were obtained from 11 PDAC patients (age 48–69, 7 male and 4 female). For preparation of human and mouse tissue samples, 10 mg PDAC or para-tumor normal tissues were immersed in 10 μl NEM Tissue Buffer (25 mM N-Ethylmaleimide, 10 mM Ammonium Formate, pH7.0) spiked with 10 μM NEM-$^{15}$N,$^{13}$C$_3$-Cysteine as interior control) and homogenized with a glass Dounce tissue homogenizer on ice. The homogenate was centrifuged at 20,000 rpm, 4 °C for 5 min and 5 μL supernatant were diluted in 45 μl NEM Tissue Buffer and used for LC-MS/MS analysis.

**GSH.** $2 \times 10^5$ cells in 12-well plate (70% confluency) were washed with ice-cold 1× PBS three times and the residual PBS was thoroughly removed. 225 μl ice-cold 0.1% formic acid and 25 μl [$^{13}$C$_2$,$^{15}$N] GSH (1 μg/ml in 0.1% formic acid) was added to the well. The cells were scrapped in to a 1.5 ml Eppendorf tube. Cells were vortexed for 15 s, snap-frozen in liquid nitrogen for 1 min and thawed in a room temperature water bath. After three freeze-thaw cycle, cells were centrifuged at $13,000 \times g$ for 3 min and the supernatant were collected for LC-MS/MS analysis.

**dNTP.** $1 \times 10^6$ cells (in 6-well plate) were washed with ice-cold PBS and residual PBS was removed. 490 μl ice-cold 80% MeOH spiked with 10 μl 100 nM [$^{13}$C$_{10}$,$^{15}$N$_5$]-dATP (646237, Sigma) internal standard were added and cells were scrapped into a 1.5 mL tube. After vortex, the lysates were centrifuged at $21,130 \times g$, 4 °C for 10 min. The supernatant was transferred to a new tube and mixed with 1 ml 50 mM $NH_4Ac$, pH 4.5 and used for subsequent SPE. Oasis WAX 1 cc, 30 mg, 30 μm cartridge (Waters, Milford, MA, USA) was used for SPE. The cartridge was activated with 1 ml methanol, and equilibrated with 1 ml ddH$_2$O prior

to sample loading. After the samples loading by gravity, the cartridge was washed by 1 mL 50 mM NH$_4$Ac, pH 4.5, and 1 mL 0.5% (25% NH$_4$OH) in methanol. The analytes were finally eluted with 1 ml buffer containing 80% methanol, 15% water, and 5% (25% NH$_4$OH). The eluent was dried by Speed-vac and reconstituted with 50 μl reconstitution solution for LC-MS/MS analysis.

**NADP+/NADPH.** $1 \times 10^6$ cells in 6-well plate were washed with ice-cold PBS and extracted with 0.4 ml 80% MeOH pre-chilled to −20 °C. The plates are transferred to −80 °C for 15 min and cells were scrapped into a 1.5 ml tube. After centrifugation at $15,000 \times g$ for 10 min at 4 °C, the supernatant was used directly of LC-MS/MS analysis.

**LC-MS/MS.** All LC-MS/MS analysis were carried out using Sciex QTRAP 6500+ mass spectrometry coupled with a Sciex EXion HPLC separation system. The multiple reaction monitoring mode (MRM) was used to analyze and quantify metabolites. All peaks were integrated and quantified by Sciex OS 3.0 software.

**Normalization.** For mouse or human tissues, the tissue weight was used for normalization. For cell culture samples, the total protein amounts from three separate replicates were determined through BCA assay and used for normalization.

### Lipase activity assay
Lipase activities were determined using Lipase Microplate Assay Kit from Absin (abs580082). Briefly, $5 \times 10^6$ cells were trypsinized and harvested through centrifugation. 1 ml Assay buffer was added to the cell pellet. After sonication, the cell lysates were centrifuged at $16,000 \times g$ at 4 °C for 40 min. The lipase activity in the supernatant was determined through a colorimetric assay according to the manufacturer's instruction.

### Aconitase activity assay
Aconitase activity in cell lysates was measured using the Aconitase Activity Assay Kit (Sigma Aldrich, MAK337) according to the manufacturer's instructions. Cytosolic aconitase (ACO1) and mitochondrial aconitase (ACO2) activities were determined by monitoring the conversion of citrate to isocitrate at 565 nm. Protein-normalized enzymatic activity was expressed as μM/min per milligram of protein.

### C11 BODIPY Lipid peroxidation assay
Cell were treated as indicated and incubated with 2 mM C11-BODIPY 581/591 (Beyotime, S0043M) for 1 h. Excess C11-BODIPY was removed by washing the cells twice with PBS. Lipid peroxidation was immediately detected by an automatic microplate spectrophotometer (Synergy H1, BioTek Instruments, Vermont, USA). The signals from both non-oxidized C11 (excitation 581 nm/emission 591 nm) and oxidized C11 (excitation 488 nm/emission 510 nm) were determined.

### FITC-cystine uptake assay
FITC-cystine uptake was performed as previously described[30]. $3 \times 10^5$ cells were plated onto 6 well plate for overnight. The cells were incubated with fresh medium containing 5 μM FITC-cystine (SCT047, Sigma). After 30 min incubation in a 37 °C CO$_2$ incubator, the cells were trypsinized and washed with ice-cold PBS (with 1% FBS) twice and used for flow cytometry analysis.

### EdU staining
Cells ($2 \times 10^5$/well) were seeded in 6-well plates and cultured overnight to allow attachment and recovery of normal growth status. Subsequently, EDU labeling, fixation, and permeabilization were performed according to the manufacturer's protocol (APExBIO, Cat# K1175). Flow cytometry data were acquired using a Beckman Coulter flow cytometer with an excitation wavelength of 488 nm, and analyzed with FlowJo software (v10.0).

### Cell proliferation assay
$1 \times 10^5$ Cells per well were seeded onto a 12-well plate in triplicate. The cell culture medium was refreshed every two days. At indicate time points, the cells were fixed with 10% formalin for 15 min and stained with 0.5 ml 10% crystal violet solution for 20 min with shaking. After staining, the cells were washed with ddH$_2$O for five times and dried overnight. The crystal violet staining was solubilized with 1 ml 10% acetic acid absorbance at OD 590 nm was determined with a plate reader.

### Colony formation assay
$3 \times 10^3$ cells were suspended in 3 mL indicated culture and seeded into 6 well plate. The medium was replaced every three days. After 10–14 days, the colonies were visualized through Crystal Violet staining.

### High throughput drug screening
Cells were seeded onto 96-well plates at an optimal density in their respective medium incubated at 37 °C in a 5% CO$_2$ environment for 16 h. Following incubation, compounds from a metabolic disease focus library (L5200, TOPSCIENCE) were added to the cells at a final concentration of 5 μM (with DMSO serving as the vehicle control). After incubation with the compounds for 48 h, cell viability was measured using the CCK-8 assay. Compounds that significantly inhibited cell viability (more than 70% inhibition) in MiaPaCa-2-CLSA cells were identified as potential hits.

### Patients tissue specimens
For NEM-cysteine experiment, de-identified tissues were obtained from 11 patients (age 48–69 years old, 7 male and 4 female). For lipid droplet and PLIN2 staining, tissues were obtained from 30 PDAC patients (age 45–72, 19 male and 11 female). For PDO cultures, tissues were obtained from 2 male PDAC patients (65–75 years old). For PDX experiments, tissues were derived from 2 patients (1 male and 1 female, age 55–75 years old). For G6PD IHC staining, tumor samples and para-tumor normal tissues were obtained from 100 patients (age 29–81, 57 male, 43 female) with PDAC treated at the Tianjin Cancer Hospital from 2014 to 2018. All patients underwent radical pancreaticoduodenectomy with R0 margins confirmed by two pathologists. None of the patients had received chemotherapy or radiotherapy at the time of sample collection. All patients were treated with systemic gemcitabine-based chemotherapy after operation for six cycles after surgery. No radiotherapy was given before or after surgery. Postoperative follow-up of patients were conducted every 3 months initially. OS was defined as the time interval from the date of surgery to that of death due to any cause or that of the last follow-up. RFS was calculated from the date of surgery to that of local recurrence or metastasis. Local recurrence or metastasis was diagnosed by radiological examination (contrast-enhanced CT or MRI scanning). All patients provided written informed consent. The study protocols were approved by the Ethics Committee of the Tianjin Cancer Institute and Hospital.

### Animal husbandry
All experimental mice were housed in a specific pathogen-free (SPF) environment. The ambient temperature was 21–23 °C, with 45% humidity and the mice had ad libitum access to water and food. The feed used was produced by Sibeifu Company (product code: SPF-F02-003). Light cycles in the mouse rooms were 12 light/ 12 dark.

### Orthotopic xenograft and bioluminescence imaging (BLI)
The orthotopic PDAC xenograft and BLI imaging were carried out following previously reported procedures[30,58,59] according to IACUC

protocols approved by the IACUC committee at the Tianjin Cancer Hospital. Briefly, Luciferase-labeled PDAC cells ($1 \times 10^5$ for KPC and Pan02 cells, $1 \times 10^6$ for MiaPaCa-2 cells) were resuspended in 20 μl Matrigel (diluted in DMEM media) and orthotopically injected into carefully exposed pancrease of 6–7 weeks old female mice (C57BL/6J for KPC and Pan02 cells, BALB/c Nude mice for MiaPaCa-2 cells). The wound was closed with suture and wound clip. Non-invasive BLI imaging was carried out using Xenogen IVIS 200 1 week after the surgery to monitored the orthotopic tumor growth. At the end of the experiment, liver were inspected for visible metastatic lessions and ex vivo BLI imaging was carried out to monitor liver metastasis.

## PDX model experiments

The experimental protocol was approved by the Ethics Committee of Tianjin Medical University Cancer Institute and Hospital, following the NIH Guide for the Care and Use of Laboratory Animals. Pathogen-free NSG mice (6–7 weeks old, female) were used for the primary tumor xenograft (PDX) model. Tumor specimens were collected from pancreatic cancer patients who underwent surgical resection, and the tissues were transplanted into 8-week-old NSG mice under a sterile environment. After three successful passages, the PDX tumor was excised and cut into 2 mm fragments and inoculated into NSG mice. Once the transplanted tumors reached 3 mm, mice were randomly assigned into 4 groups (6 mice per group) to receive different treatments. The treatment groups included: (A) vehicle control (saline), (B) mFOLFIRINOX (Oxaliplatin: 10 mg/kg, intravenous injection once weekly; 5-FU: 10 mg/kg, intraperitoneal injection once weekly; Irinotecan: 5 mg/kg, intraperitoneal injection once weekly), as previously described[60], (C) Lomitapide treatment at 10 mg/kg via daily oral gavage, and (D) combination therapy with mFOLFIRINOX plus Lomitapide. Tumor size was measured weekly using calipers, and the tumor volume was calculated with the formula: volume = length × width × width/2. At the end of the experiment, tumors were harvested from the mice for further processing and analysis.

For PET-CT scan, mice were fasted for 6 h prior to the experiment but had free access to water. Before the injection of the imaging agent, the mice were weighed, and $^{18}$F-FDG was administered via the tail vein at a dosage of 5 μCi/g, with the injection time recorded. One hour after the tail vein injection of the 18F-FDG tracer, a micro-PET/CT scan was performed. The mice were anesthetized using isoflurane gas and positioned prone on the small animal PET/CT bed, with infrared beams used to center the scanning field. Static imaging was then conducted, starting with the PET scan followed by the CT scan, for a total duration of approximately 10 min. The CT and PET images were transmitted to the IRIS workstation for image reconstruction and fusion.The equipment we used is the French Inviscan small animal imaging system, IRIS PET/CT, and the $^{18}$F-FDG radioactive tracer was provided by the Molecular Imaging and Nuclear Medicine Department of Tianjin Cancer Hospital.

## KPC mouse model

KPC mice (C57BL/6Smoc-*Kras*$^{em4(LSL-G12D)}$; *Trp53*$^{em4(R172H)}$; *Pdx1-cre* mice, 8 weeks old, male) were purchased from Shanghai Model Organisms Center. The development of pancreatic cancer in these mice were monitored by ultrasound scans (Canon Aplio i800, Japan). Mice with tumor volumes of 20–60 mm³ were randomly assigned into four groups: (A) vehicle control (saline), (B) mFOLFIRINOX, (C) Lomitapide, and (D) combination therapy with mFOLFIRINOX plus Lomitapide and treated as described above. The treatments continued for 4 weeks, during which the tumor volume was monitored biweekly via ultrasound scans. The survival of mice continued to be monitored after stopping the treatment.

## IHC analysis

Sections of formalin-fixed, paraffin-embedded tumors were subjected to IHC analysis for G6PD and PGD as using a DAB (3,3′-diaminobenzidine) substrate kit (Maixin) according to the manufacturer's instructions. Sections were incubated with anti-G6PD (sc-373886, Santa Cruz Biotechnology, at 1:200 dilution) or anti-PGD (sc-398977, Santa Cruz Biotechnology, at 1:200 dilution), anti-ATF4 (10835-1-AP, Proteintech, at 1:200 dilution) or anti-PLIN2 (Abcam, ab52356, at 1:100 dilution) antibodies overnight at 4 °C. After wash, the sections were stained with a secondary antibody for 30 min at room temperature. PBS was substituted for each primary antibody as a negative control. Five random fields were examined under a light microscope. Immunoreactivity was scored according to the estimated percentage of positive tumor cells as described previously. IHC staining results were blindly and independently performed by two pathologists who were blinded to the clinical data. Staining intensity was scored 0 (negative), 1 (low), 2 (medium), and 3 (high). The H-score was calculated according to the formula: H-score = (0 × percentage of 1+ cells) + (1 × percentage of 1+ cells) + (2 × percentage of 2+ cells) + (3 × percentage of 3+ cells), yielding a total score ranging from 0 to 300. Expression levels were classified based on the H-score as follows: 0 (negative, −), 1–100 (weak expression, +), 101–200 (moderate expression, ++), and 201–300 (strong expression, +++). Antibodies used in IHC staining can be found in Supplementary Table S2.

## Lipid droplets staining

For the staining of LD in tissues, we subjected fresh cancer/cancer-adjacent tissues from patients or mouse tumor fragments to rapid freezing, followed by embedding in OCT compound and sectioning. The sections were stored at −20 °C. Prior to staining, the sections were gradually thawed at 4 °C for 20 min, then permeabilized with ice-cold acetone for 20 min. Afterward, sections were blocked with 3% BSA for 30 min. Subsequently, BODIPY™ 493/503 NHS ester (succinimidyl ester) dye was added at a final concentration of 5 μM for the staining of adiposomes. The sections were incubated at room temperature for 60 min, washed three times with PBS, and counterstained with DAPI. For quantitation, The average LD numbers per cell from 5 random fields (20x objective) were counted.

For lipid droplet staining in cells, $3 \times 10^5$ cells were seeded onto a 3.5 cm glass bottom cell culture dish (50% confluency). The glass bottom was coated with 0.2% gelatin as previously described[61]. Cells were stained with 1.5 μl LipidSpot™-488 dye diluted in 1 ml cell culture medium in a 37 °C CO₂ incubator for 30 min. After staining, the cells were washed with 1 × PBS and fixed with 4% paraformaldehyde for 10 min, washed with 1 × PBS twice and counterstained with DAPI. The LD were imaged with Leica SP8 laser scanning confocal microscope equipped with an 63 × oil immersion objective.

## Thin layer chromatography (TLC)

$5 \times 10^6$ cells in a 10 cm dish were washed with 1 × PBS twice and harvested with a cell scrapper. The cell pellets were wash with 1 × PBS twice and the lipids were extracted with 750 μL methanol:choloroform (4:1, v/v). After centrifugation at 13,300 rpm for 5 min at room temperature, the supernatant was mixed with 600 μl 0.1% aqueous acetic acid and 300 μl chloroform and vortexed for 10 s. The mixture was centrifuged at 13,300 rpm for 5 min and the lower organic phase was transferred to a glass vassel and dried under nitrogen. The dried lipids were reconstituted in chloroform and loaded onto Silica gel 60 TLC plate (1.11845.0001, Merck). The TLC plates were developed using hexane/diethyl ether/acetic acid 60:40:1 (v/v/v). The lipids were visualized using iodine vapor and imaged with a scanner.

## Oleate-Alkyne tracing

$2 \times 10^6$ PDAC cells in a 6 cm dish were incubated with media containing 25 μM Oleic acid-Alkyne and 1% fatty acid free BSA for 30 min. To determine the effects of lomitapide, cells were pre-treated with lomitapide at various concentrations for 2 h prior to Oleic acid-alkyne labeling. After labeling, lipids were extracted and reacted with 3-azido-

7-hydroxyconmarin as previously described[62]. The Oleic acid-alkyne labeled lipids were spotted onto a TLC plate (Merk, 1118450001) and developed with hexane/diethyl ether/acetic acid 60:40:1 (v/v/v).

## DGAT activity assay

DGAT activity assay was performed using 1,2 dioleoyl-sn-glycerol (DOG, D0138, Sigma) and NBD-palmitoyl CoA (810705 P, Avanti Polar Lipids) as substrate and crude miccrosomal membrane isolated from mouse liver following a previously reported protocol[46]. 5 μg liver membrane (in 50 μl buffer containing 50 mM Tris-HCl, pH 7.6, 250 mM sucrose) was incubated with 150 μl reaction master mix (20 μl 1 M Tris-HCL, pH 7.6, 1 μl 1 M MgCl$_2$, 10 μl 4 mM DOG, 10 μl 12.5 mg/ml FFA-free BSA, 10 μl 500 μM NBD-palmitoyl CoA and 99 μl ddH$_2$O) with or without lomitapide, mixed by vortexing and incubated at 37 °C for 10 min with shaking. The reaction was terminated by the addition of 750 μl methano/chloroform 4:1 v/v followed by vortexing. After $17,000 \times g$ centrifugation at room temperature for 5 min, the 400 μl lower organic phase was transferred to a glass tube, dried with SpeedVac. After reconstituion in chloroform, the NBD-labeled lipids were resolved through TLC as previously described.

## Metabolomics and lipidomics screening

Metabolomics screening was performed through LC-MS/MS as previously described[63,64]. The MSEA analysis was performed using Metaboanalyst 5.0. Untargeted lipidomics screening was performed with assistance from Dr. John Asara (BIDMC Mass Spectrometry Core) following previously their published protocols[65]. Briefly, $7 \times 10^6$ cells were washed with PBS and harvested with a cell scraper in a 12 ml glass tube. After centrifugation ($1000 \times g$, 5 min at room temperature), the cells were resuspended in 200 μl PBS. 10% cells were saved for protein assay and normalization. The remaining cells were extracted with 4 ml chloroform:methanol (2:1, v/v). After 30 min incubation on a orbital shaker at room temperature, 400 μl ddH$_2$O was added and the samples were vortexed for 1 min. The samples were allowed to stand for 10 min and centrifuged at $2200 \times g$ for 15 min. The lower phase was collected and dried under a nitrogen stream and used for untarged metabolomics screening using a relative quantitative positive/negative ion switching method as previously described[65].

Targeted quantitative lipidomics assay was performed with assistance from Metware Bio (Woburn, MA). For lomitapide treatment, CLSA-25 MiaPaCa-2 cells were treated with 1 μM lomitapide for 24 h. For mFOLFIRINOX treatment, $1 \times 10^7$ naïve MiaPaCa-2 cells were treated with mFOLFIRINOX cocktail consists of 5-FU (8 μM), Oxaliplatin (8 μM), and SN-38 (4 μM) for 24 h. For PANC-1 targeted lipidomics, $1 \times 10^7$ CLSA-25 or naïve control cells were used. Cells were harvested using a cell scrapper. The cell pellets were resuspended in 100 μl ddH$_2$O containing proteinase inhibitors. 50 μl cell suspensions were used for lipid extraction, while the remaining samples were used for protein assay and the total protein amounts in each sample was used for normalization. The data acquisition instruments consisted of Ultra Performance Liquid Chromatography (NexeraLC-40, Shimadzu) equipped with Thermo Accucore™C30 (2.6 μm, 2.1 mm × 100 mm i.d.) column and tandem mass spectrometry (MS/MS) (Triple Quad 6500+, AB/SCIEX). Liquid phase conditions: Mobile phase: A phase was acetonitrile /water (60/40, V/V) (0.1% formic acid added, 10 mmol/L ammonium formate); B phase was acetonitrile/Isopropyl alcohol (10/90, V/V) (0.1% formic acid added, 10 mmol/L ammonium formate); Gradient program: 80:20 (V/V) at 0 min, 70:30 (V/V) at 2 min, 40:60 (V/V) at 4 min, 15:85 (V/V) at 9 min, 10:90 (V/V) at 14 min, 5:95 (V/V) at 15.5 min, 5:95 (V/V) at 17.3 min, 80:20 (V/V) at 17.5 min, 80:20 (V/V) at 20 min; Flow rate: 0.35 ml/min; Column temperature: 45 °C; Injection volume: 2 μL. Mass spectrometry conditions:

LIT and triple quadrupole (QQQ) scans were acquired on a triple quadrupole-linear ion trap mass spectrometer (QTRAP), QTRAP® 6500 + LC-MS/MS System, equipped with an ESI Turbo Ion-Spray interface,operating in positive and negative ion mode and controlled by Analyst 1.6.3 software (Sciex). The ESI source operation parameters were as following: ion source, turbo spray; source temperature 500 °C; ion spray voltage (IS) 5500 V (Positive), −4500 V (Negative); Ion source gas 1 (GS1), gas 2 (GS2), curtain gas (CUR) were set at 45, 55, and 35 psi, respectively. Instrument tuning and mass calibration were performed with 10 and 100 μmol/L polypropylene glycol solutions in QQQ and LIT modes, respectively. QQQ scans were acquired as MRM experiments with collision gas (nitrogen) set to 5 psi. DP and CE for individual MRM transitions was done with further DP and CE optimization. A specific set of MRM transitions were monitored for each period according to the lipids eluted within this period.

## $^{13}C_6$, $^{15}N_2$-cystine tracing

Cystine isotopologue tracing was performed following a previously described protocol[66] with modifications. Cystine, methionine, and glutamine free DMEM media (Gibco, 21013-024) were supplemented with 400 μM (for control group) or 50 μM (for CLSA-25 group) $^{13}C_3$, $^{15}N$-cysteine, 100 μM methionine, 4 mM glutamine and 10% dialyzed FBS and stored at 4 °C for 48 h for the oxidation of cysteine to cystine. $7 \times 10^6$ control or CLSA-25 MiaPaCa-2 cells were pre-conditioned in the same medium containing $^{12}C,^{14}N$-cystine for 12 h. After pre-conditioning, cells were labeled with their respective media containing isotopologue cystine for indicated time (2, 4, or 24 h). The metabolites were extracted with 2 ml 80% MeOH. After centrifugation, the supernatant was dried under nitrogen flow and analyzed through LC-MS/MS as previously described[67].

## $^{13}C_6$ -glucose tracing to glycolysis and PPP/nucleotide metabolites

$^{13}C_6$-glucose tracing was performed as we described previously with modifications[63,68]. $7 \times 10^6$ control or CLSA-25 MiaPaCa-2 cells were incubated with DMEM medium (Life Technology, D98002710) containing 200 μM (for control group) or 25 μM cystine (for CLSA-25 group), 10 mM $^{13}C_6$-glucose, and 10% dialyzed FBS for indicated time (from 0 to 360 min). After extraction with 80% methanol and centrifugation, the supernatant was collected and dried using SpeedVac and re-dissolved in 50 μL of methanol/water (50/50) for LC-MS/MS analysis as previously described[68].

## $^{13}C_5$, $^{15}N_2$ -glutamine tracing for de novo fatty acid synthesis

$1 \times 10^6$ PANC-1 cells on a 6 cm dish was washed with PBS and incubated with isotopic labeling media containing 4 mM $^{13}C_5$, $^{15}N_2$ -glutamine containing 200 μM (for control cells) or 25 μM (for CLSA-25 cells) L-Cys$_2$ and dialyzed FBS. After 24-h incubation, cells were washed with 1 × PBS twice and scrapped into a 1.5 ml tube on ice. The cells were centrifuged at $500 \times g$ at 4 °C for 3 min and the cell pellets were vortexed in 1 ml 50% methanol with 0.05 M HCl for 10 s. 500 μl choloroform was added to each tube and vortexed vigorously to extract lipids. After $16,000 \times g$ centrifugation for 5 min, the lower organic phase was transferred into a glass vial. The upper aqueous phase was extracted with choloroform again and the organic phase was combined and dried under nitrogen flow. The dried lipids were reconstituted in 90% methanol solution containing 0.3 M KOH and incubated in a 2 ml glass vial at 80 °C for 1 h. After saponification, the 100 μl formic acid was added to the reaction and the mixture was extracted with 900 μl hexane twice. The hexane extracts were combined and dried under nitrogen flow. After reconstitution in 50 μl methanol/chloroform 4:1 v/v, the saponified fatty acids were used for mass spectrometry analysis. The LC/MS system consisted of a Dionex Ultimate 3000 XRS pump (Thermo Scientific, San Jose, CA), a Dionex Ultimate 3000 XRS open autosampler (Thermo Scientific, San Jose, CA), a Dionex Ultimate 3000 RS column compartment (Thermo Scientific, San Jose, CA), and an Exactive Plus orbitrap mass spectrometer (Thermo Fisher Scientific, San Jose, CA). Data was collected using Xcalibur 3.0.63 software (Thermo Scientific, San Jose, CA) and analyzed

using FreeStyle 1.8.63.0 software (Thermo Scientific, San Jose, CA). A binary gradient of solvent A: 97/3 water/methanol with 10 mM tributylamine and 15 mM acetic acid (pH 4.5) and solvent B: 100% methanol was as follows: 80 to 99% B from 0 to 20 min, remaining at 99% B from 20 to 40 min, from 99% B to 80% B to 41 min, and remaining steady at 80% B to 50 min. The separation was performed on a Luna C8(2) reversed-phase column (150 × 2.0 mm, 3 μm particle size, 100 Å pore size, Phenomenex, Torrance, CA). The flow rate was 200 μL/min, the autosampler temperature was 5 °C, injection volume 10 μL, and column temperature 25 °C. The mass spectrometer was operated in negative mode. The electrospray settings were set as follows sheath gas flow rate 45 (arbitrary units), auxiliary gas flow rate 10 (arbitrary units), sweep gas flow rate 2 (arbitrary units), spray voltage 2.5 kV, capillary temperature 250 °C, S-lens RF level 50, and auxiliary gas temperature 400 °C. The mass spectrometer resolution was set to 140,000 and the automatic gain control was set to $3 \times 10^6$ with a maximum injection time of 100 ms. The scan range was 200–400 m/z from 0 to 20 min and 300 to 575 m/z from 20 to 50 min.

### [1-¹³C]-L-Serine tracing to transsulfuration

L-Cystine and L-Serine Free DMEM high glucose Media was supplemented with 400 μM [1-¹³C]-L-Serine, and 400 μM (for control cells) or 50 μM (for CLSA-25 cells) L-Cysteine and 10% FBS. [1-¹³C]-L-Serine and L-Cysteine were added to the Media 48 h before test. Cell were preconditioned in medium containing 10% FBS and ¹²C-serine for 12 h in 10 cm dish ($7 \times 10^6$ cells per dish). Cells were washed twice with 1 × PBS and incubated with their respective media containing 400 μM [1-¹³C]-L-Serine for 4 h. The cells were then washed with ice-cold PBS for three times and lysed in 2 ml 80% MeOH in −80 °C freezer for 15 min. The cells were harvested through scraping. After $17,000 \times g$ centrifugation at 4 °C for 20 min, the supernatant was transferred to a new tube and dried with nitrogen flow and used for LC-MS/MS analysis.

### [U-¹³C]-OA tracing

$8 \times 10^6$ MiaPaCa-2 cells were pretreated with 2 μM lomitapide or vehicle (DMSO) in regular growth media (DMEM plus 10% FBS) for 2 h. After 2-h pretreatment, cells were incubated with growth media supplemented with 10 μM with vehicle control or 2 μM lomitapide, respectively, for 6 h. The cells were then washed with ice-cold PBS and harvested using a cell scraper. 1/10 of the cells were used for protein assay (BCA method) and normalization. The remaining cells were lysed with 1 ml 50% MeOH, 0.05 M HCl and vortexed for 10 s. 0.5 ml chloroform was added and vortexed vigorously to extract lipids. After $16,000 \times g$ centrifugation for 5 min at 4 °C, the lower organic phase was added to a glass vial. The upper aqueous phase was extracted with 0.5 ml chloroform again and the two organic phase extractions were combined and used for SPE separation using STRATA NH2 column (500 mg, 55 μm, 70 A, 8B-S009-HBJ, Phenomenex) following a previously reported protocol[43,69]. The column was preactivated with 4 ml hexane, followed by 4 ml chloroform/isopropanol (2:1 v/v) and then 4 ml hexane again. The MiaPaCa-2 lipid extracts were loaded. Neutral lipids, FFA, and polar lipids were eluted using 4 ml chloroform/isopropanol (2:1 v/v), 8 ml diethyl ether/acetic acid (100:2, v/v), and 4 ml methanol, respectively. After elution, the lipids were dried under nitrogen flow, saponified and used for LC-HRMS analysis (see **¹³C₅, ¹⁵N₂-glutamine tracing for de novo fatty acid synthesis**).

### Matrix-assisted laser desorption/ionization (MALDI)-mass spectrometry imaging (MSI) of cysteine and cystine in mouse pancreas

Mouse pancreas samples (approximately 3 × 3 mm) were harvested from 8-weeks old KFC mice (Kras^LSL-G12D; Trp53^fl/fl; Pdx1-Cre) and immersed immediately in isopentane placed on top of dry ice. The frozen samples were cryosectioned at a thickness of 7 μm using a Leica CM1950 cryostat. Tissue sections were mounted onto indium tin oxide (ITO)-coated glass slides for MALDI-MSI and onto standard glass slides for histological analysis. A MALDI matrix solution consisting of 1,5-diaminonaphthalene (5.6 mg/mL in 5% HCl and 50% ethanol) was applied to the ITO-mounted sections using the HTX M3+ automated sprayer under the following conditions: flow rate of 20 μL/min, 25 passes, 10 psi liquid nitrogen sheath gas, and a spraying velocity of 1200 mm/min. MALDI-MSI was conducted using a Thermo Q Exactive HF-X Orbitrap mass spectrometer integrated with a UV-laser MALDI source (Spectroglyph LLC)[70]. Data acquisition was performed in negative ion mode across an m/z range of 100–1000 with a spatial resolution of 20 μm. Adjacent serial sections of the mouse pancreas were fixed in 4% formaldehyde post-sectioning and stained with hematoxylin and eosin (H&E) stain, processed in University of Texas Health Science Center at San Antonio (UTHSA). Autofluorescence (AF) images of pre-MSI sections and brightfield images of H&E-stained sections were acquired using a Zeiss Axioscan 7 digital slide scanner equipped with a 20 × objective lens. Please refer to the protocols.io for detailed procedures at Zhang et al.[71].

Cysteine and cystine annotations from MSI were executed using METASPACE, a software with false discovery rate control[72]. To confirm molecular identity, tandem mass spectrometry (MALDI-MS/MS) was performed on frozen mouse pancreas sections targeting the cysteine and cystine precursor ion. The resulting fragment ion m/z values were verified by Human Metabolome Database spectral library[73]. The spatial distribution of cysteine and cystine in frozen mouse pancreas sections was analyzed using Bruker SCiLS Lab software. MALDI-MSI datasets for each sample were imported into SCiLS, and corresponding pre-MSI AF images were registered to the total ion current (TIC) image using a two-point alignment method. H&E-stained images from adjacent serial sections were subsequently overlaid onto the AF images. Regions of interest (ROI, Normal, PanIN & ADM, Tumor, and Lymphocytes) from mouse pancreas were defined based on H&E-stained images by a pancreatic cancer pathologist. Ion intensity data corresponding to cysteine (m/z 120.0125) and cystine (m/z 239.0166) were extracted for each pixel within the ROI following TIC normalization. Mean ion intensity values for cysteine and cystine were calculated by averaging pixel-level intensities within the ROI.

### Drug treatment synergy analysis

The Zero Interaction Potency (ZIP) score analysis was performed using SynergyFinder[74] to determine the synergistic effects between Lomitapide and mFOLFIRINOX treatments. ZIP score >10 indicates synergistic effects between the two treatments.

### Material availability

All unique/stable reagents generated in this study are available from the lead contact with a completed materials transfer agreement.

### Statistics and reproducibility

Statistical data analysis was performed using Graphpad Prism 10. The continuous outcome variables were compared using two-tailed two-sample t-test (paired or unpaired). For non-continuous variables, Mann–Whitney U test was used. For multiple comparisons, one-way ANOVA with Dunnett's or Sidak's multiple comparison test were used. For tumor growth curves and cell proliferation curves, two-way ANOVA with Tukey's multiple comparison test were used. Categorical variables were analyzed using Fisher's exact test of Chi square test. Survival analysis was performed using Kaplan–Meier's method. No statistical method was used to predetermine sample size. No data were excluded from the analyses. For cell-based experiments, western blotting, immunostaining, and FACS, data collection was conducted blindly. Measurement for cell viability, FACS, photo capture, and histological analysis were performed by different individuals who were blinded to the experimental groups. Mass spectrometry analysis was blinded prior to analysis.

## Ethical considerations

All animal experiments were performed with approval from the Animal Ethical and Welfare Committee (AEWC) at the Tianjin Medical University Cancer Institute and Hospital (Approval No.: 2023086). The following humane endpoints were used for mouse euthanasia: a single tumor volume of ≥2000 mm³; persistent tumor surface ulceration and infection that showed no improvement after local care (e.g., sterile disinfection); deterioration of the mice's general condition, such as a body weight loss of ≥20% within one week accompanied by emaciation, rough and unkempt fur, and loss of skin elasticity; refusal to eat or drink for 24 consecutive hours with signs of dehydration (e.g., sunken eye sockets, dry nasal mucosa, and cold limbs); and abnormal movement and behavior, including persistent huddling and inactivity, unresponsiveness to stimuli, or the presence of ataxia (unsteady gait, stumbling) and paralysis.

The research using de-identified human samples (organoids, PDXs, and patient tissue specimens) complies with all relevant ethical regulations. In accordance with the requirements of the Ethics Committee of Tianjin Medical University Cancer Institute and Hospital, China, and the recognized ethical guidelines of the Declaration of Helsinki, all patients provided written informed consent for the use of their specimens and disease-related information in future research (Approval No.: EK20220153). The patients are not compensated.

## Reporting summary

Further information on research design is available in the Nature Portfolio Reporting Summary linked to this article.

## Data availability

Metabolomics and lipidomics data are being deposited to National Metabolomics Data Repository (NMDR) (accession: PR002249; https://doi.org/10.21228/M8XC02) and are publicly available as of the date of publication. Source data are provided with this paper.

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

## Acknowledgements

We would like to thank the Flow Cytometry Core (RRID: SCR_021134) and Mass Spectrometry Core (RRID: SCR_017831) and Dr. Yuan-Wan Sun at Penn State College of Medicine for their technical support and Dr. Xianming Deng for critically reading the manuscript. We would like to thank the Core Facility of the Research Center of Basic Medical Sciences at Tianjin Medical University for assistance with drug screening service. The research in S. Yang's lab is supported by grants from the NIH (R01CA256911, R01CA233844) and the DOD (HT9425-23-1-0348). The research in J.Hao and X.Wang's lab is supported by grants from National Natural Science Foundation of China (grants 82430057, 82273362, 82472973). Dr. Pankaj Singh is supported by grants from the NIH (R01CA163649, R01CA270234, R01CA256911, and U54CA274329).

## Author contributions

Y.L. and Z.L. performed most of the experiments with assistance from Q.L., B.N., M.T., A.E.S., M.W., C.M., G.S., B.F., Y.S., R.R.T., T.Z., Y.X., B.Q., S.Z., and G.Z. D.S., A.E.S., and K.-M.C. developed the mass spectrometry methods for metabolite analysis. Z.L., C.M., G.S., B.F., Y.S., T.Z., Y.X., X.W., and J.H. validated the upregulation of OxPPP and LD in PDAC patient tissues. Z.L. performed the drug screening and validation with

assistance from C.M., G.S., B.F., and Y.S. Y.L. performed polysome profiling experiments and translational regulation analysis with assistance from S.R.K. and Y.D. G.H.R. and P.K.S. analyzed the glucose flux to glycolysis and OxPPP pathways through $^{13}$C tracing. S.Z. and G.Z. performed MS imaging experiments to analyze the spatial distribution of Cys and $Cys_2$ in PDAC tissues. Y.L., Z.L., X.W., J.H., and S.Y. interpret the data. S.Y. wrote the manuscript with inputs from all authors.

## Competing interests

Shengyu Yang, Jihui Hao, Xiuchao Wang, Yunzhan Li, and Zekun Li are named inventors of a pending patent application. Other authors declare no conflict of interest.

## Additional information

[1]Department of Cell and Biological Systems, Penn State Cancer Institute, Penn State College of Medicine, Hershey, PA, USA. [2]Penn State Cancer Institute, Penn State College of Medicine, Hershey, PA, USA. [3]Department of Pancreatic Cancer, Tianjin Medical University Cancer Institute and Hospital, Tianjin, China. [4]National Clinical Research Center for Cancer, Key Laboratory of Cancer Prevention and Therapy, Tianjin, China. [5]Department of Neuroscience Experimental Therapeutics, Mass Spectrometry Core Facilities, Penn State College of Medicine, Hershey, PA, USA. [6]Mass Spectrometry Core Facilities, Penn State College of Medicine, Hershey, PA, USA. [7]The Huck Institutes of the Life Sciences, The Pennsylvania State University, University Park, PA, USA. [8]Metabolomics Core Facility, The Pennsylvania State University, University Park, PA, USA. [9]The Center for Precision Medicine, Division of Nephrology, Department of Medicine, University of Texas Health San Antonio, San Antonio, TX, USA. [10]Department of Biochemistry and Molecular Biology, Penn State College of Medicine, Hershey, PA, USA. [11]School of Chemical Engineering, Tianjin University, Tianjin, China. [12]Center for Life Science, School of Life Sciences, Yunnan University, Kunming, China. [13]Department of Oncology Science, University of Oklahoma Health Sciences Center, Oklahoma City, OK, USA. [14]These authors contributed equally: Yunzhan Li, Zekun Li. ✉e-mail: wangxiuchao@tjmuch.com; haojihui@tjmuch.com; sxy99@psu.edu

