## [Transparent Peer Review file · Nature Communications]

Adaptation to cystine limitation stress confers a targetable lipid metabolism vulnerability in pancreatic ductal adenocarcinoma

Corresponding Author: Dr Shengyu Yang

Version 0:

Reviewer comments:

Reviewer #1

(Remarks to the Author)

This manuscript by Li Y, Li Z et al., titled "Adaptation to Cystine Limitation Stress Confers a Targetable Lipid Metabolism Vulnerability in Pancreatic Ductal Adenocarcinoma (PDAC)," explores the connection between cysteine limitation adaptation and lipid reprogramming in PDAC.

While many individual aspects such as cyst(e)ine depletion, ferroptosis, oxPPP, and lipid reprogramming are already recognized features of PDAC and might initially seem to lack novelty, the innovative aspect of this study lies in its integration of these elements and the proposal of a new drug combination (LOMITAPIDE) with mFOLFIRINOX chemotherapy. The authors also used a multifaceted approach in multiple PDAC models.

Therefore, I consider this study to be of significant relevance to the scientific community and the readership of Nature Communications. However, several major and minor concerns, listed below, need to be addressed before it can be considered for publication.

Major Concerns

1. The starting point of CLSA seems mainly to be attributed to adaptation to a decreased uptake of cystine, as pointed out with the use of IKE and by comparison with the para-tumor content of cyst(e)ine. However, it's unclear whether in this context there is an impaired SLC7A11 gene expression over time or rather due to usage of cysteine by other pathways beyond GSH. The authors should confirm the most salient results of the manuscript using inducible loss- and gain-of-function of SLC7A11. Is there a loss of SLC7A11 in PDAC patients that can account for a worse prognosis or resistance to chemotherapy? Or is there a role of the pancreas microenvironment in the sequestration of cyst(e)ine?
2. Even though mass spectrometry was performed in tumors and para-tumor samples, Figure 1 is missing the Cys2 quantification and the ratio. It would be more relevant and convincing to show Cys and Cys2 spatial distribution in tumor and non-tumor areas via MALDI-MS imaging with a corresponding H&E staining.
3. It would be beneficial to have images of CLSA occurring over time in Figure 1C and Supplementary Figure 1, not just at the endpoint. Is CLSA reversible or a permanent event?
4. C11-BODIPY should be expressed as ratiometric values (Green/Red) and not as %high. Please revise Supplementary Figure 1 accordingly. Additionally, these methods and others are missing in the methods section.
5. As cyst(e)ine and GPX4 operate on the same axis of ferroptosis regulation, it is not surprising that CLSA PDAC cells become co-resistant to the inhibition of both. What about other pathways regulating ferroptosis? It's known that cancer cells resistant to the cyst(e)ine/GPX4 pathway can be vulnerable to inhibition of other independent pathways such as FSP1 and DHFR, which are also tightly connected with oxPPP and nucleotide metabolism/synthesis. The authors should show whether these contribute to the phenotype of CLSA in PDAC.
6. From PDAC orthotopic model experiments, CLSA PDAC cells seem to maintain the more aggressive phenotype shown in vitro. This again suggests a cell-autonomous and permanent adaptation of the PDAC cells even when there is plenty of cyst(e)ine in the pancreas microenvironment. This point is connected to comment number 1 and needs to be dissected with orthogonal loss- and gain-of-function approaches.
7. Supplementary Figure 1: Can cyst(e)ine and Trp deprivation be rescued by BSA? The text says 2-4 weeks, the legend says 2 weeks, and panel J says days instead of weeks—what is the timepoint? The authors should consolidate the results and explain why certain results are missing or are not consistent.
8. In Figure 2A, the authors find a decrease in the M+4 cysteine derived from labeled cystine; however, the M+0 fraction of

cysteine and all other metabolites in Supplementary Figure 2C is increased. The authors suggested briefly that this could be due to additional sources of cysteine, but none of these possibilities were tested or shown. This is particularly relevant as some PDAC cell lines used in this study have a baseline good expression of the transsulfuration pathway genes. Even though PDAC and other cancers seem to rely mainly on cyst(e)ine uptake, the upregulation of the transsulfuration pathway can contribute to the CLSA phenotype. This hypothesis is also corroborated by the ATF4 upregulation (a regulator of the CSE gene) and by the absolute absence of labeling of HTAU and TAU. Authors need to show a C13-serine tracing in vitro and in vivo, with and without CLSA. Also, protein levels of the main components of the de novo synthesis and uptake of cyst(e)ine should be shown in all conditions. Refer to PMID: 36862034 for guidelines regarding this point.

9. The authors state that CLSA cells have a decrease in the labeled alanine coming from cysteine as a byproduct of the Fe-S cluster pathway. This is linked to a decreased activity of both ACO1 and ACO2 activity (these methods are missing), as well as to Fe-S formation and impaired mitochondrial respiration. Two points require attention: What's the impact on the mitochondrial mass? And what's the role of iron metabolism in this phenotype? Key components of the Fe-S pathways such as ISCU and CISD1 are not taken into consideration. It is also unclear why the authors chose H3 as a housekeeping control in Figure 2D.

10. MSEA analysis in Figure 3A should show the significant sets that are up and down in CLSA conditions. Also, a heatmap or volcano plot for all the significant metabolites at steady-state should be reported at least in the Supplementary section.

11. G6PD mRNA levels need to be assessed in multiple cell lines. Also, it's unclear whether the CHX assay was run in CLSA or control cells. siRNA-mediated knockdown of G6PD should be confirmed with inducible shRNA.

12. The authors state that "G6PD overexpression increases ferroptosis resistance." However, they are just targeting the xCT/GPX4 pathway, which is only one of the pathways. This comment relates to comment #5: The authors need to check other ferroptosis pathways. Also, they should explain why G6PDⁱ or KD induce only a partial rescue of the CLSA-induced phenotype (this is valid also for results in Figure 5J, K).

13. Lipidome analysis in Figure 5 should include a more thorough analysis, including double bonds and carbon length analysis for all the relevant classes. Also, PC, PE, and PS significantly change in CLSA, but they are not taken into consideration. As these lipids can be both synthesized and stored in LD, they can play a role in the phenotype that is neglected now. Also, they should use an LD-enriched fraction and perform lipidomics on that to prove the relevant differentially regulated lipids are confined there. The same criticism is valid for lipidomics in Figures 6 and 7.

14. Whenever LDs are accumulating, this could indicate also a decreased lipolysis, which is not taken into consideration. Specific lipase activities should be measured in all conditions to consider the effect of lipolysis in the CLSA phenotype. Lipase inhibitors should also be used to test this hypothesis. The same is valid for the in vivo experiments in Figures 6 and 7.

15. Panel L in Figure 5 needs a corresponding H&E to evaluate the result. It would be more informative to see tumor and para-tumor areas together on the same slide.

16. The 800-drug screen focused on metabolism is nice. However, the lack of a full table and the raw data renders the evaluation of these results difficult. In other words, the authors really need to prove in the most unbiased way that Lomitapide is the best candidate drug to be used. Also, the plausible target of Lomitapide, MTTP, seems to be missing in PDAC cells—which cells are we looking at in Supplementary Figure 6B? Control or CLSA? Also, by implying that Lomitapide targets DGAT, the authors point to potential off-target effects of the drug. There are several other lipid synthesis inhibitors (e.g., FASN, SCD inhibitors) that are available and FDA-approved, which produce a phenotype like Lomitapide but are proven to be very on-target and less promiscuous than Lomitapide. The authors should consider using other inhibitors of the same pathways to corroborate their conclusions.

17. Lipidomics of Lomitapide-treated versus control is again neglecting other relevant classes such as PE, PC, and LPC that significantly change. In Figure 6G, PL are not even represented, which seems odd. Refer to comment 13.

18. Regarding Supplementary Figure 6J, K—Microsomal analysis should be done using microsomes isolated from isogenic control and CLSA cells +/- treatments or DGAT.

19. In all in vivo studies, including the one in Figure 6, I miss the confirmation of the lipidomic profile seen in vitro. This is especially relevant because the cells are implanted in a cyst(e)ine-replete environment, and we don't know if the phenotype is maintained post-implantation and treatment. Also, it would be relevant to have confirmation that the drug is on-target and that it reaches the tumors. What's the overall toxicity and the effect on the general systemic metabolism in animals?

20. Fig 6 J-K It would be cool to use a fluorescent-azide and use click chemistry to show incorporation of OA-alkyne into LD stained with a different color.

21. General lipid incorporation can be altered in ctrl vs CLSA conditions. Authors need to test this aspect. Also, a general MS analysis with a traceable OA should be used to measure the incorporation of OA in lipids other than TG.

22. Figure 7- Synergy of Lom +FOLFIRINOX should be showed in a synergy matrix.

23. In all the in vivo experiments, what is the contribution of ferroptosis? Is ferroptosis the eventual mechanism triggering tumor regression/inhibition?

Minor Concerns

24. In general, the results discussion is at points abrupt and should be elaborated more (e.g. "These findings implicated the involvement of other metabolic pathways in addition to OxPPP and G6PD"—which ones?). The authors should include possible limitations too. The discussion is mostly a summary of the results without much elaboration and integration with existing literature. Most of the discussion should be used in the results instead. General statements such "according to the expression patterns of", or "modulates", render the readability a bit confusing and jeopardize the results. I advise to revise the text and to better guide the reader through the figures with tangible variables that are the same in both text and figure. The authors should also make sure they use a proper scientific terminology (e.g. "expression" refers to genes and not to proteins)

25. Methods, figure legends and figures are sloppy in several points and need to be extensively revised and implemented. Some methods are missing, some others are lacking details (e.g normalization, timepoints etc). Figure legends do not provide a comprehensive explanation of the panels and miss important details, hindering a proper evaluation of the results. Some figures and legends are mislabeled. Quantification is missing in multiple panels (e.g LD quantification). Labels of

graphs are sometimes missing or not immediate and complicate their interpretation (e.g. Log 2 FC of what?; Fold Change of what?; Expression in PDAC cells (which ones? Ctrl or CLSA))

Reviewer #2

(Remarks to the Author)

The present study investigates how pancreatic ductal adenocarcinoma (PDAC) cells adapt to chronic cystine limitation—an environmental stress common in the nutrient-deprived PDAC microenvironment. The authors show that cells which adapt to cystine limitation (termed “CLSA” cells) not only maintain glutathione synthesis despite low cystine levels but also undergo metabolic reprogramming. Key features of this reprogramming include a translational upregulation of the oxidative pentose phosphate pathway (OxPPP) (notably increased levels of G6PD, PGD, and PGLS), which fuels enhanced nucleotide synthesis and lipogenesis. The increased lipogenesis leads to higher triacylglyceride (TG) production and lipid droplet (LD) formation—a process that helps the cells mitigate lipotoxicity while promoting proliferation and ferroptosis resistance. A drug screen identified lomitapide, an FDA-approved lipid metabolism drug, as a potent inhibitor of TG synthesis (likely via DGAT inhibition) in these adapted cells. Lomitapide not only induces lipotoxicity in CLSA cells but also sensitises PDAC cells and patient-derived models to chemotherapy (FOLFIRINOX), leading to improved anti-tumor efficacy in xenograft and genetically engineered mouse models. Overall, the study was well conducted and of broad interest. I only have a few remarks the authors might wish to address:

- 1) The authors report the activation of ATF4 at 24 hours upon cysteine limitation. Is the basal activation of ATF4 maintained in the CLSA cells? Expanding on this, does this generate a long-standing dependence on ATF4 activity, and ultimately, is the constitutive activation of ATF4 the causative event leading to increased dependency on TG synthesis? Finally, does the ATF4 activation also observed in the tumor samples? This could add molecular depth to some of the observations reported by the authors.
- 2) The aspect describing the increase ferroptosis resistance of CLSA is not worked out in detail. A better characterization, including evaluating the expression of GPX4, SLC7A11, FSP1, for example, would be of interest. Along these lines, to what extent is the PUFA/MUFA ratio changed in the different phospholipid classes. Also, can the inhibition of LD restore the ferroptosis sensitivity in these cells? A recent preprint (<https://www.biorxiv.org/content/10.1101/2025.01.06.631537v1>) has hinted at a role of FSP1 in preserving TG and inhibiting ferroptosis within lipid droplets, perhaps this is worth discussing in line with the present observations.

Minor points:

- Line 127 and Line 129, starting with “Although..” are duplicated.
- The word “reprograming” is used in several places; the standard spelling is “reprogramming.”
- Line 121 - “ferroptosis-like cell death in PDAC cells in in 72 hours” contains a duplicate “in.” It should read “in 72 hours.”
- Line 448 - In the Discussion section, “miroenvironment” appears instead of “microenvironment.”

Reviewer #3

(Remarks to the Author)

The article by Li et al. studies the adaptation to Cystine Limitation Stress (CLS) in PDAC. By using cell lines, patient-derived organoids, patient samples, and diverse mouse models, the authors demonstrate that adaptation to CLS promotes tumor growth by activating the oxidative pentose phosphate pathway via increased translation. CLSA induces lipid reprogramming and de novo lipid synthesis, resulting in triacylglyceride (TG) synthesis and accumulation in lipid droplets to mitigate lipotoxicity. The compound Lomitapide inhibits TG synthesis, interfering with lipid reprogramming and enhancing FOLFIRINOX chemotherapy both in vitro and in vivo.

I have thoroughly reviewed this manuscript and must commend the authors on the excellent quality of work presented. The research is insightful, well-structured, and contributes significantly to the field. The methodology is sound, and the conclusions drawn are both logical and impactful.

However, there are a few minor corrections and suggestions that I would like to point out:

1. Is LD staining increased in patients with high G6PD expression? These data would significantly strengthen the story, fully supporting the in vivo/in vitro data presented here on the connection between high oxPPP and increased de novo lipid synthesis.
2. Considering that the patients used for survival analysis (Table S1) were treated with gemcitabine-based therapies, it would be important to determine if lomitapide sensitizes to gemcitabine or gemcitabine/abraxane on top of mFOLFIRINOX, at least using in vitro assays.
3. There are a few typographical errors throughout the text.
4. The statistical analysis methodology must be justified and included in the Materials and Methods section.

Version 1:

Reviewer comments:

Reviewer #1

(Remarks to the Author)

I commend the authors for such a thorough and extensive revision!

Well done! In my opinion, the manuscript is now suitable for publication in Nature Communications

A few minor comments/suggestions:

- 1) I found both the BSA supplementation and the OA incorporation results very interesting and I definitely suggest to investigate these mechanisms in the future studies.
- 2) Lomitapide seems to be potent regardless of CLSA, so I would probably highlight this in the discussion
- 3) I spotted a few typos: lines 409 ("Taking" instead of "Taken"); 433 (Flg); 458, 576 ("Lipidomics" instead of "Lipidome")
- 4) There are 2 reference sections-all references should be together
- 5) Suppl Fig. 9 panel E misses the color legend

Reviewer #2

(Remarks to the Author)

The authors have sufficiently addressed my concerns.

I would only avoid stating that the resistance cells upregulate Coq biosynthesis proteins since the effect observed is only marginal.

Reviewer #3

(Remarks to the Author)

No further comments

We would like to thank the three reviewers for their many constructive critiques and valuable suggestions. We have performed extensive new experiments and thoroughly revised the manuscript to address to the comments, as detailed in the following point-by-point response.

Point-by-point response to reviewer comments

Reviewer 1

This manuscript by Li Y, Li Z et al., titled "Adaptation to Cystine Limitation Stress Confers a Targetable Lipid Metabolism Vulnerability in Pancreatic Ductal Adenocarcinoma (PDAC)," explores the connection between cysteine limitation adaptation and lipid reprogramming in PDAC.

While many individual aspects such as cyst(e)ine depletion, ferroptosis, oxPPP, and lipid reprogramming are already recognized features of PDAC and might initially seem to lack novelty, the innovative aspect of this study lies in its integration of these elements and the proposal of a new drug combination (LOMITAPIDE) with mFOLFIRINOX chemotherapy. The authors also used a multifaceted approach in multiple PDAC models.

Therefore, I consider this study to be of significant relevance to the scientific community and the readership of Nature Communications. However, several major and minor concerns, listed below, need to be addressed before it can be considered for publication.

Reviewer #1 (Remarks to the Author):

Major Concerns

1. The starting point of CLSA seems mainly to be attributed to adaptation to a decreased uptake of cystine, as pointed out with the use of IKE and by comparison with the para-tumor content of cyst(e)ine. However, it's unclear whether in this context there is an impaired SLC7A11 gene expression over time or rather due to usage of cysteine by other pathways beyond GSH. The authors should confirm the most salient results of the manuscript using inducible loss- and gain-of-function of SLC7A11. Is there a loss of SLC7A11 in PDAC patients that can account for a worse prognosis or resistance to chemotherapy? Or is there a role of the pancreas microenvironment in the sequestration of cyst(e)ine?

Response: Our western blotting results showed that treatment with 25 uM modestly reduced SLC7A11 protein levels at 7 days and after 4 weeks adaptation (Fig. S1I). However, despite reduction in SLC7A11 protein levels, the CLSA PDAC cells increased FITC-cystine uptake activities by 20-50% (Fig. S1J). Consistent with this observation, CLSA cells had much higher (approximately 4-fold higher) intracellular cysteine when cultured in cystine replete medium (200 uM cystine) (Fig. S1K) according to LC-MS/MS

quantitation of NEM-Cysteine. The mechanism underlying the increased cystine uptake is unclear.

Following the suggestion of this reviewer, we used tet-on shRNA to knockdown SLC7A11 in PDAC cells (knockdown efficiency shown in Fig. S4D). As shown in Fig. S4D, although acute knockdown of SLC7A11 (doxycycline, 5 days) had no effect on the protein levels of OxPPP enzymes, the protein levels of G6PD, PGD and PGLS increased after 4 weeks dox-induction (100ng/ml). Chronic SLC7A11 knockdown (4 weeks) also induced adaptation that promoted cell proliferation and ferroptosis resistance (Fig. 1H and S1L). Taken together, our results showed that CLS (induced by either depletion of environmental cystine or by genetic depletion of its transporter) could induce adaptations that activate OxPPP and promote cell proliferation, ferroptosis resistance.

To determine whether there is a loss of SLC7A11 in PDAC, we used IHC to compare the expression of SLC7A11 in 60 PDAC tissues and paired para-tumor normal pancreatic tissues (Fig. R1A). Our data showed that there was no difference in SLC7A11 in PDAC vs normal tissues. Analysis of survival data from TCGA PDAC dataset (kmplot.com) also showed that there was no significant correlation between SLC7A11 transcript levels and patient survival (Fig. R1B).

Figure R1, A, representative IHC staining micrograph (left) and quantitation (right) showing the expression of SLC7A11 in PDAC tissues and paired normal tissues from PDAC patients. B, the correlation between SLC7A11 mRNA transcript levels and PDAC patient overall survival in the TCGA PDAC patient cohort. Data were from kmplot.com. C, representative IHC staining of CDO1 in human liver tissues, PDAC tissues and paired normal pancreas tissues.

To determine whether the tumor microenvironment plays a role in cystine sequestration, we used IHC to examine the expression of CDO1, the enzyme responsible for cystine catabolism. As shown in Figure R1C, although normal pancreatic islet cells had weak CDO1 staining, the pancreatic acinar cells in normal tissues, cancer cells and stromal cells in PDAC tissues were negative for CDO1 expression. Taken together, our data suggested that the cysteine/cystine depletion in the tumor microenvironment was likely not engaged in the active cysteine/cystine catabolism.

The lower intracellular cysteine in PDAC tissues could be due to a combination of factors such as increased consumption (due to hyperactive redox flux in cancer cells) and decreased supply (due to desmoplasia and defective vasculatures). In our MS imaging experiment, tumor infiltrating lymphocytes in PDAC tumor tissues had higher

levels of Cys than cancer cells (Fig. 1C, Fig. S1A). It remained unclear whether these tumor infiltrating lymphocytes in PDAC tissues compete with cancer cells for cystine to induce CLS.

2. Even though mass spectrometry was performed in tumors and para-tumor samples, Figure 1 is missing the Cys2 quantification and the ratio. It would be more relevant and convincing to show Cys and Cys2 spatial distribution in tumor and non-tumor areas via MALDI-MS imaging with a corresponding H&E staining.

Response: New Cys2 and Cys/Cys2 ratios data are provided in Fig. 1B. We have also performed MS imaging to determine the Cys, Cys2, Cys/Cys2 ratios in regions corresponding to PDAC tissues, normal tissues and PanIn/ADM. Our data (Fig. 1C and Fig. S1A) showed that Cys and Cys/Cys2 ratios were decreased in PanIN/ADM and further decreased in PDAC regions when compared to normal pancreatic tissue regions. Cys2 levels, however, was not significantly different between normal, PanIN and PDAC in MS imaging experiment.

3. It would be beneficial to have images of CLSA occurring over time in Figure 1C and Supplementary Figure 1, not just at the endpoint. Is CLSA reversible or a permanent event?

Response: To obtain the colony formation assay results in supplementary Figure 1 and Fig. 1C, it is necessary to grow the cells undisturbed for approximately two weeks to allow each single cell to form colonies. At earlier time points the colony will be too small for visualization. To determine the intermediate effect of CLSA on PDAC cell proliferation, we used EdU to label the proliferating cells (cells in S phase) after 25 uM Cystine treatment for 0-4 weeks. As shown in Fig. 1E, CLS-25 treatment for 3 days markedly reduced EdU positive cells by approximately 80% in PDAC cells. The proportion of EdU positive cells gradually recover after 1 week culture in 25 uM Cystine medium and exceed the control cells by week 3- week 4 (Fig. 1E).

To determine whether CLSA phenotypes are transient or stable, we cultured CLSA cells in cystine replete medium (200 uM) for 4 weeks. As shown in Fig. S1R-S1S and S4E, after 4 week culture in the Cys2-replete media, the cell proliferation, ferroptosis resistance and G6PD expression in CLSP MiaPaCa-2 cells remained elevated when compared to naïve control MiaPaCa-2, which suggested that CLSA is a very stable phenotype, and that the maintenance of these phenotypes didn't require the continuous presence of CLS.

4. C11-BODIPY should be expressed as ratiometric values (Green/Red) and not

as %high. Please revise Supplementary Figure 1 accordingly. Additionally, these methods and others are missing in the methods section.

Response: C11-BODIPY results in the original manuscript were determined using flow cytometry using flow cytometry as we described in a previous publication¹. Following the suggestion of this reviewer, we measured C11-BODIPY green-red ratios using fluorescence plate reader and the results were used to replace the flow cytometry results in this revised manuscript (Fig. S1D).

5. As cyst(e)ine and GPX4 operate on the same axis of ferroptosis regulation, it is not surprising that CLSA PDAC cells become co-resistant to the inhibition of both. What about other pathways regulating ferroptosis? It's known that cancer cells resistant to the cyst(e)ine/GPX4 pathway can be vulnerable to inhibition of other independent pathways such as FSP1i and DHFR, which are also tightly connected with oxPPP and nucleotide metabolism/synthesis. The authors should show whether these contribute to the phenotype of CLSA in PDAC.

Response: As suggested by this reviewer, we tested the effect of CLSA on sensitivities to FSP1 inhibitor iFSP1 and DHFR inhibitor (methotrexate). As shown in Fig. S1M and S1N, CLSA decreased the IC50 of FSP1i from 16.5 uM and 7.9 uM in control MiaPaCa-2 cells and PANC-1 to 1.2 uM and 0.7 uM in their respective CLSA-25 cells (Fig. S1M). CLSA also increased sensitivities to DHFRi, but to a smaller extent (Fig. S1N). Therefore, as suggested by this reviewer, while CLSA increased resistance to ferroptosis inducers targeting cystine/GPX4 pathway, CLSA cells were more vulnerable iFSP1/DHFRi inhibition.

6. From PDAC orthotopic model experiments, CLSA PDAC cells seem to maintain the more aggressive phenotype shown in vitro. This again suggests a cell-autonomous and permanent adaptation of the PDAC cells even when there is plenty of cyst(e)ine in the pancreas microenvironment. This point is connected to comment number 1 and needs to be dissected with orthogonal loss- and gain-of-function approaches.

Response: Indeed, as suggested by this reviewer, CLSA phenotypes are stable and not dependent on the continuous presence of CLS, as indicated by responses to comment 3.

7. Supplementary Figure 1: Can cyst(e)ine and Trp deprivation be rescued by BSA? The text says 2-4 weeks, the legend says 2 weeks, and panel J says days

instead of weeks—what is the timepoint? The authors should consolidate the results and explain why certain results are missing or are not consistent.

Response: 2-4 weeks was the time required for PDAC cells to adopt stable adaptation in their nutrient-limited media. The 2 weeks in the legend of colony formation assays were the time needed for colony formation. In these assays naïve control cells or adapted cells were plated in their respective media and cultured undisturbed for 2 weeks to allow the formation of colonies from single cells. The panel J was a cell proliferation assay where naïve or adapted cells were plated on day 0 and their cell proliferation were evaluated over the course of days. Therefore the inconsistency in time course indicated by this reviewer was due to the nature of different experiments and assays.

New BSA rescue data are included in Fig. S1F and S1W. Supplementation with 5% BSA not only had no rescue effect on MiaPaCa2 colony formation in cystine limitation

condition, but

paradoxically further inhibited colony growth. In Trp-limited media PDAC cells was able to maintain cell proliferation for 2 weeks. Similar to 25 μ M cystine limitation, including 5% BSA in the Trp-limited media inhibited MiaPaCa-2 cell colony formation (Fig. R2). After 2 weeks, MiaPaCa-2 cells grew in Trp-limited media stopped proliferating. Supplementation with 5% BSA had no rescue effect (Fig. S1W).

We must admit that we are puzzled by the inhibitory effects of BSA supplementation in Cystine- and Trp-limited media. However, these experiments have been repeated by three different individuals (Qin Li, Yunzhan Li and Zekun Li) in three different locations (Penn State College of Medicine, Tianjin Cancer Hospital and China Pharmaceutical University) using different batches of BSA and cell culture media and similar results

were achieved. Therefore, the inhibitory effect of 5% BSA supplementation in Cys2 and Try limited conditions are highly repeatable and the underlying mechanisms will need to be investigated in the future.

8. In Figure 2A, the authors find a decrease in the M+4 cysteine derived from labeled cystine; however, the M+0 fraction of cysteine and all other metabolites in Supplementary Figure 2C is increased. The authors suggested briefly that this could be due to additional sources of cysteine, but none of these possibilities were tested or shown. This is particularly relevant as some PDAC cell lines used in this study have a baseline good expression of the transsulfuration pathway genes. Even though PDAC and other cancers seem to rely mainly on cyst(e)ine uptake, the upregulation of the transsulfuration pathway can contribute to the CLSA phenotype. This hypothesis is also corroborated by the ATF4 upregulation (a regulator of the CSE gene) and by the absolute absence of labeling of HTAU and TAU. Authors need to show a C13-serine tracing in vitro and in vivo, with and without CLSA. Also, protein levels of the main components of the de novo synthesis and uptake of cyst(e)ine should be shown in all conditions. Refer to PMID: 36862034 for guidelines regarding this point.

Response: Gina DeNicola's group reported in PMID: 36862034 that cancer cells (including PDAC cells) had little to no detectable de novo synthesis of cysteine through transsulfuration pathway in cell culture or in vivo, despite expression of CBS and CSE. We confirmed the expression of CBS and CSE in PDAC cells and CSE expression was elevated in CLSA MiaPaCa-2 cells (Fig. S2E). However, when C13-serine was used as tracer, there was only less than 1% labeling of cysteine in control MiaPaCa-2 cells and no detectable labeling in CLSA cells (Fig. S2F). We observed a modest increase in the labeling of cystathionine by 13C-serine (Fig. S2F) in CLSA cells (from 27% to 35%). Although there was 20-30% labeling of GSH, fractional labeling decreased in CLSA cells (15%) when compared to control cells (32 %). The lack of cysteine labeling suggested that the GSH labeling by C13-serine was likely due to the conversion of C13-serine into C13-glycine. Our results were consistent with results reported by the DeNicola group and suggested that the transsulfuration pathway is unlikely to be a significant source of cysteine in control or CLSA PDAC cells.

It has been previously reported that nutrient limitation promotes macropinocytosis. Therefore, we used fluorescent-labeled dextran to examine the effects of CLSA on micropinocytosis. As shown in Fig.S2G-S2H, we observed significant increase in micropinocytosis in CLSA PDAC cells. Therefore, cysteine uptake from micropinocytosis could potentially account for the increased in M+0 cysteine and other sulfur metabolites in the stable state tracing data in Fig. S2D.

9. The authors state that CLSA cells have a decrease in the labeled alanine coming from cysteine as a byproduct of the Fe-S cluster pathway. This is linked to a decreased activity of both ACO1 and ACO2 activity (these methods are missing), as well as to Fe-S formation and impaired mitochondrial respiration. Two points require attention: What's the impact on the mitochondrial mass? And what's the role of iron metabolism in this phenotype? Key components of the Fe-S pathways such as ISCU and CISD1 are not taken into consideration. It is also unclear why the authors chose H3 as a housekeeping control in Figure 2D.

Response: Flow cytometry analysis of Mitotracker Green staining in control and CLSA PDAC cells indicated that there was no significant difference in mitochondrial mass between naïve and adapted PDAC cells (Fig. S2B). Western blotting analysis of TFRC, IRP1, IRP2, ISCU and NFS1 showed upregulation of TFRC and IRP2, which indicated upregulation of the iron starvation response (Fig. 2F). Interestingly, ISCU and NFS1, two core components of the ISC assembly complex, were also downregulated in CLSA cells (Fig. 2F). These data are consistent with the critical roles of Fe-S in the regulation of iron starvation response. Furthermore, the downregulation of ISCU and NFS1 in CLSA cells would likely contribute to the suppression of Fe-S synthesis in CLSA cells.

Both tubulin and histone 3 were used as loading control in the original Fig. 2D Western blotting. We have now included Tubulin in addition to Histone H3 in the revised figure (Fig. 2E).

Aconitase activity assay method is now included in the revised Materials and Methods.

10. MSEA analysis in Figure 3A should show the significant sets that are up and down in CLSA conditions. Also, a heatmap or volcano plot for all the significant metabolites at steady-state should be reported at least in the Supplementary section.

Response: MSEA analysis has been performed on upregulated metabolites and downregulated metabolites (Fig. S3B, S3D). In addition, volcano plot of all the significantly changed metabolites were included in Fig. S3A, S3B.

11. G6PD mRNA levels need to be assessed in multiple cell lines. Also, it's unclear whether the CHX assay was run in CLSA or control cells. siRNA-mediated knockdown of G6PD should be confirmed with inducible shRNA.

Response: We confirmed that, similar to MiaPaCa-2 cells, CLSA had little effect on the G6PD, PGLS and PGD mRNA levels in CLSA PANC1 cells (Fig. S2G). The CHX assay was performed in naïve control MiaPaCa-2 and new data in CLSA25 MiaPaCa-2 cells showed similar results (Fig. S2H). We also confirmed that inducible shRNA depletion of

G6PD abrogated CLSA-mediated ferroptosis resistance and cell proliferation (Fig.S2M-S2O).

12. The authors state that “G6PD overexpression increases ferroptosis resistance.” However, they are just targeting the xCT/GPX4 pathway, which is only one of the pathways. This comment relates to comment #5: The authors need to check other ferroptosis pathways. Also, they should explain why G6PDI or KD induce only a partial rescue of the CLSA-induced phenotype (this is valid also for results in Figure 5J, K).

Response: As suggested by this reviewer, we tested the effects of G6PD overexpression on iFSP1. As shown in Fig. S4R, ectopic expression of G6PD in naïve MiaPaCa-2 cells and PANC-1 cells also increase resistance to iFSP1 (IC50 increased from 4.8 uM to 9.6 uM in PANC-1 and 11.1 uM to 30.9 uM in MiaPaCa-2).

The partial rescue effect of G6PDI1 or G6PD knockdown in some assays are likely due to metabolic changes in other pathways. We have included the following paragraph in the discussion:

The metabolic reprogramming in CLSA PDAC was in part due to suppression of Fe-S synthesis and translational activation of the OxPPP. While genetic or pharmacological inhibition of G6PD abrogated the CLSA-mediated tumor cell proliferation and ferroptosis resistance, ectopic expression of G6PD could only partially recapitulate some of the CLSA phenotypes (e.g. ferroptosis resistance, lipogenesis), but not others (e.g. cell proliferation). These findings implicated the involvement of other metabolic pathways in addition to OxPPP during CLSA. A common adaptation response to chronic amino acid deprivation stress is the compensatory activation of the mTORC1, which promotes both nucleotide and lipid synthesis. The potential role of mTORC1 in CLSA will need to be further investigated in the future.

13. Lipidome analysis in Figure 5 should include a more thorough analysis, including double bonds and carbon length analysis for all the relevant classes. Also, PC, PE, and PS significantly change in CLSA, but they are not taken into consideration. As these lipids can be both synthesized and stored in LD, they can play a role in the phenotype that is neglected now. Also, they should use an LD-enriched fraction and perform lipidomics on that to prove the relevant differentially regulated lipids are confined there. The same criticism is valid for lipidomics in Figures 6 and 7.

Response: The lipidomics data from MiaPaCa2 control and CLSA cells in Figure 5 was untargeted / non-quantitative and not suitable for detailed analysis. Therefore, we performed new targeted/quantitative lipidomics experiment using PANC1 control and CLSA cells (Fig. 6A). Using these new data, we performed double bonds and carbon

length analysis in major lipid classes (FFA, PC, PE, PI, PS, PG, TG) and data are shown in Fig. S7. Our data showed that like MiaPaCa2 cells, TG was also prominently upregulated in CLSA cells. Interestingly, PC and Cer, two lipid classes downregulated in CLSA MiaPaCa2 cells were upregulated in CLSA PANC-1, which indicated that CLSA-induced lipidomic changes of different lipid species could be context dependent.

Similar analysis have been performed for targeted lipidomics data in the original figures 6 (lomitapide treatment) and 7 (FOLFIRINOX treatment).

Also, they should use an LD-enriched fraction and perform lipidomics on that to prove the relevant differentially regulated lipids are confined there.

Response: This appeared to be a misunderstanding of our view. We believed that the lipidomics reprogramming in CLSA cells are global and not limited to TG or lipid droplets. Our *de novo* FA synthesis assay using ¹³C-glutamine as tracer was examining FA synthesis in the whole lipidome, not limited to TG or LD. In fact, it is well established that FA flux among different lipid classes is intricately connected. Blocking FA flux into one lipid subclass (e.g. TG synthesis) would likely result in increasing FA flux to other subclass (e.g. acyl-CoA, acyl-carnitine, PL etc). The reason that we focused on TG and LD in this manuscript was because our mechanistic studies showed that lomitapide inhibits TG synthesis and LD formation. We agree with this reviewer, lipidomics changes in other lipid classes could have important implications for CLSA phenotypes and PDAC progression. However, due to immense complexity of lipidomes, the context dependent changes in different lipid subclasses (as shown in the new lipidomics screening in PANC1 cells) and the scope of the current study, we choose to focus on the global lipogenesis in general instead of changes in different subclasses.

To better reflect our rationale to focus on TG and LD, we have reorganized Figure 5 and 6. Most of the data in the original Figure 6 are now presented in Figure 5 and the original Figure 5 are now Figure 6 in the revised manuscript.

14. Whenever LDs are accumulating, this could indicate also a decreased lipolysis, which is not taken into consideration. Specific lipase activities should be measured in all conditions to consider the effect of lipolysis in the CLSA phenotype. Lipase inhibitors should also be used to test this hypothesis. The same is valid for the in vivo experiments in Figures 6 and 7.

Response: As suggested by this reviewer, we performed ATGL lipase activity assay using cell lysates from control and CLSA MiaPaCa-2 and PANC1 cells. Our new data showed that the lipase activity in control and CLSA PDAC cells were not statistically significant (Fig. S8A). ATGL inhibitor itself had little cytotoxicity in either control or CLSA

PDAC cells (Fig. S10F). When combined with gemcitabine, ATGL inhibitor **decreased** (instead of increased) the sensitivities of gemcitabine chemotherapy in control and CLSA PDAC cells (Fig. S10G). Therefore, our data suggested that the accumulation of TG and LD in CLSA PDAC cells were not due to inhibition of lipolysis and ATGL inhibition would not an effective approach to increase chemosensitivity. Given these observations, we didn't test the ATGLi in animal models.

15. Panel L in Figure 5 needs a corresponding H&E to evaluate the result. It would be more informative to see tumor and para-tumor areas together on the same slide.

Response: Panel L has been replaced with data showing H&E staining and LD staining on serial sections of PDAC tissues and para-tumor normal tissues (revised Fig. 6L). Frozen PDAC tissues and para-tumor normal tissues in our tissue banks are small in sizes and it is rare to have both normal and cancer tissues on the same block. However, we were able to identify large paraffin blocks that contain both normal and cancer tissues. Paraffin blocks are not suitable for staining with lipid droplet fluorescent dye (BODIPY 493/503 or LipidSpot), due to the use of organic solvents in sample preparation. Therefore, we used immunofluorescence staining of PLIN2 as a marker for LD. Our PLIN2 staining results showed that LD was upregulated in PDAC tissues when compared to para-tumor normal tissues (Fig. S8H).

16. The 800-drug screen focused on metabolism is nice. However, the lack of a full table and the raw data renders the evaluation of these results difficult. In other words, the authors really need to prove in the most unbiased way that Lomitapide is the best candidate drug to be used. Also, the plausible target of Lomitapide, MTPP, seems to be missing in PDAC cells—which cells are we looking at in Supplementary Figure 6B? Control or CLSA? Also, by implying that Lomitapide targets DGAT, the authors point to potential off-target effects of the drug. There are several other lipid synthesis inhibitors (e.g., FASN, SCD inhibitors) that are available and FDA-approved, which produce a phenotype like Lomitapide but are proven to be very on-target and less promiscuous than Lomitapide. The authors should consider using other inhibitors of the same pathways to corroborate their conclusions.

Response: The summary of the 800 drug screening results were shown in Fig 5A and we have now included the data for this panel in the corresponding source data file. As shown in the source data table, lomitapide was the number 3 most effective drugs in the drug screening and the number 1 among the FDA approved drugs. To the best of our knowledge, there is no FDA approved FASN or SCD1 inhibitors for the treatment of

cancer or other diseases. Orlistat is an FDA-approved gastric and pancreatic lipase inhibitor that was reported to have potential FASN inhibitory activities. However, like lomitapide, this is also a repurposed drug instead of a FASN-specific inhibitor. In our manuscript, we did identify through drug screening and validate other FDA-approved lipid metabolism drugs such as mevastatin and darapladib (Fig. 5B). However these drugs were less effective than lomitapide.

As suggested by this reviewer we tested FASN and SCD1 inhibitors Denifanstat and A939572 (both were tested in clinical trials) in MiaPaCa2 cells. As shown in Fig. S5B, CLSA PDAC cells were more sensitive to denifastat (IC 50: 66.5 uM and 14.6 uM for naïve and CLSA MiaPaCA2 cells; 30.7 uM and 7.5 uM for naïve and CLSA PANC1 cells). A939572 had little effect on PDAC cell viability even at 100 uM and the IC50s were not determined.

We would also like to respectfully point out that although inhibitions of lipid synthesis (by targeting FASN and SCD1) could potentially inhibit TG synthesis and LD formation, the mechanisms of action are different from DGAT inhibitors. Lipid synthesis inhibitors would not be effective in targeting TG/LD due to elevated lipid/FA uptake or lipolysis. On the other hand, inhibitors targeting DGAT would be effective in targeting TG/LD from de novo FA synthesis, lipid uptake or lipolysis alike.

17. Lipidomics of Lomitapide-treated versus control is again neglecting other relevant classes such as PE, PC, and LPC that significantly change. In Figure 6G, PL are not even represented, which seems odd. Refer to comment 13.

Response: In the original Figure 6G, we didn't include PL in the volcano plot because we intended to emphasize increases in toxic lipid species (FA and CAR). Including PL would mask FA and CAR on the plot. Following the suggestion of this reviewer, we have replotted the data, to include PC, PE, LPC and other major lipid classes in several different volcano plots (Fig. 5G, Fig. S5F-S5I). In addition, we have performed carbon length and double bond analysis (Fig. S5J).

--

18. Regarding Supplementary Figure 6J, K—Microsomal analysis should be done using microsomes isolated from isogenic control and CLSA cells +/- treatments or DGAT.

Response: New experiments using microsomes isolated from control and CLSA cells were performed. Similar to liver microsome experiments, lomitapide could inhibit DGAT activities in PDAC cell microsome and pre-incubation reduced or abrogated the inhibitory activities of lomitapide toward DGAT activity (Fig. S6H).

19. In all in vivo studies, including the one in Figure 6, I miss the confirmation of the lipidomic profile seen in vitro. This is especially relevant because the cells are implanted in a cyst(e)ine-replete environment, and we don't know if the phenotype is maintained post-implantation and treatment. Also, it would be relevant to have confirmation that the drug is on-target and that it reaches the tumors. What's the overall toxicity and the effect on the general systemic metabolism in animals?

Response: Our new immunofluorescence staining using PLIN2 antibody confirmed that CLSA MiaPaCa2 xenograft tumors had more LD than control tumors and lomitapide treatment reduced LD staining (Fig. S6C-S6D). We used PLIN2 staining because these tumor samples were preserved via FFPE and not suitable for LipidSpot or BODIPY 493/503 staining. In the previous submission, we showed LipidSpot LD staining in tumor tissues harvested from PDX models and our results showed that chemotherapy increased LD staining, which could be inhibited by lomitapide treatment (Fig. S10K-S10L). Our new PLIN2 staining in FFPE samples from KPC model showed similar results to PDX experiments (Fig. 7M).

As an FDA-approved drug, the toxicity and side effects of lomitapide have been extensively characterized in pre-clinical animal studies and clinical trials. The real-world evaluations of the safety and efficacy of long-term lomitapide therapy among HoFH patients have also been extensively documented in the literature²⁻⁴. Overall, long term administration lomitapide therapy was found to be safe among patients and the dose that we used in animal studies were relevant to the dose in patients. Most common adverse events were manageable, mild to moderate gastrointestinal side effects (e.g. diarrhea, nausea etc). Another common side effect is steatosis and increase in liver inflammation; however, most of the liver side effects were observed after long-term lomitapide therapy (19 months or longer) and progression to liver fibrosis was very rare. General effects of lomitapide on systemic metabolism among patients included lower LDL-c and fat accumulation in the liver, due to its inhibition of MTTP.

In our animal experiments we observed no weight loss from lomitapide single treatment; when combined with FOLFIRINOX, the combination group had no additional weight loss when compared to FOLFIRINOX group (Fig. S10J). We discuss the side effects of lomitapide in the limitation section of the revised manuscript. Future development of lomitapide-based DGAT inhibitors without MTTP inhibitory effects would be desirable.

20. Fig 6 J-K It would be cool to use a fluorescent-azide and use click chemistry to show incorporation of OA-alkyne into LD stained with a different color.

Response: We did try OA-alkyne labeling in PDAC cells and unfortunately, we were not able to achieve good labeling of LD. This is likely because TG localized to the hydrophobic core of LD and poorly accessible to Cu^{2+} , which is needed to catalyze the click-chemistry reaction. The click-chemistry reaction for extracted lipids (in TLC experiments) happened in organic solvents using $[\text{acetonitrile}]_4\text{CuBF}_4$ as catalyst. Such conditions would destroy LD structure and therefore not suitable for microscopy.

Figure R3, Control and CLSA-25 MiaPaCa-2 cells were incubated with 10 μM $[\text{U-}^{18}\text{C}]$ OA for 6 hours. The lipids were extracted and separated into FFA, neutral lipids and polar lipids using a SPE column. The lipids were saponified and used for the analysis of fractional OA labeling through LC-HRMS.

21. General lipid incorporation can be altered in ctrl vs CLSA conditions. Authors need to test this aspect. Also, a general MS analysis with a traceable OA should be used to measure the incorporation of OA in lipids other than TG.

Response: Due to the immense complexity of lipidome, it is technically challenging to trace FA incorporation into thousands of different lipid species. Therefore, we used solid phase extraction to separate the lipids into three main categories: neutral lipids

(including TG, DG, acyl-ceramides etc), free fatty acids and polar lipids (including phospholipids and lysophospholipids). Using $[\text{U-}^{13}\text{C}]$ -OA as tracer, we observed increased OA incorporation into polar lipids in CLSA MiaPaCa-2, but not to FFA or neutral lipids. We are including this data in this response letter for the information of this reviewer (Figure R3). The increased incorporation into polar lipids could be due to the elevated cell proliferation (and thus increase in membrane biogenesis). We would like to further investigate this hypothesis in future studies. However, if this reviewer felt it was necessary, we could include the data in the revised manuscript.

Using SPE separation and $[\text{U-}^{13}\text{C}]$ -OA tracing, we also demonstrated in new experiments that lomitapide inhibited OA incorporation into neutral lipids, with no significant effects on incorporation into polar lipids or FFA (Fig. 6F).

22. Figure 7- Synergy of Lom +FOLFIRINOX should be showed in a synergy matrix.

Response: The synergy of lomitapide plus FOLFIRINOX data were The Zero Interaction Potency (ZIP) score analysis was performed using SynergyFinder⁵ to determine the synergistic effects between Lomitapide and mFOLFIRINOX treatments. Data are shown in Fig. S10C.

23. In all the in vivo experiments, what is the contribution of ferroptosis? Is ferroptosis the eventual mechanism triggering tumor regression/inhibition?

Response: As shown in Fig.S5C, pan-caspase inhibitor ZVAD and RIPK1 inhibitor Necrostatin-1 could partially rescue, lomitapide-induced cell death, at least in CLSA cell. However, ferrostatin-1 had no effect. C11-Bodipy staining in cell culture (Fig. S5D) and and 4HNE staining of lomitapide-treated PDX-1 tumor sections (Fig. R4) showed that lomitapide treatment had little effect on lipid peroxidation in vitro or in vivo. Taken together our data showed that lomitapide likely induced cell death through a combination of apoptosis and necroptosis, but not ferroptosis. With that said, we have new experiments showing that when combined with RSL-3, lomitapide could increase sensitivity to RSL-3 induced ferroptosis and overcame CLSA-mediated ferroptosis resistance (Fig. S6E).

Minor Concerns

24. In general, the results discussion is at points abrupt and should be elaborated more (e.g. “These findings implicated the involvement of other metabolic pathways in addition to OxPPP and G6PD”-which ones?). The authors should include possible limitations too.

The discussion is mostly a summary of the results without much elaboration and integration with existing literature. Most of the discussion should be used in the results instead. General statements such “according to the expression patterns of”, or “modulates”, render the readability a bit confusing and jeopardize the results. I advise to revise the text and to better guide the reader though the figures with tangible variables that are the same in both text and figure. The

authors should also make sure they use a proper scientific terminology (e.g. “expression” refers to genes and not to proteins)

Response: We have revised the discussion section to incorporate the constructive critiques of this reviewer, including providing more detailed discussion of other potential metabolic pathways, adding a new limitation sections, removing or shortening the parts describing or summarizing results, among others. Results section has also been revised to include tangible variables such as IC50 and numerical values. In addition, all references to “protein expression” were revised to “protein abundance” or “protein levels”.

25. Methods, figure legends and figures are sloppy in several points and need to be extensively revised and implemented. Some methods are missing, some others are lacking details (e.g normalization, timepoints etc). Figure legends do not provide a comprehensive explanation of the panels and miss important details, hindering a proper evaluation of the results. Some figures and legends are mislabeled. Quantification is missing in multiple panels (e.g LD quantification). Labels of graphs are sometimes missing or not immediate and complicate their interpretation (e.g. Log 2 FC of what?; Fold Change of what?; Expression in PDAC cells (which ones? Ctrl or CLSA)

Response: Methods and Figure legends have been revised to include additional details (such as normalization and timepoints) and to improve clarity. All the LD staining have been quantified. The figures and legends have been checked multiple rounds to correct for mistakes and to improve clarity.

Reviewer #2 (Remarks to the Author):

The present study investigates how pancreatic ductal adenocarcinoma (PDAC) cells adapt to chronic cystine limitation—an environmental stress common in the nutrient-deprived PDAC microenvironment. The authors show that cells which adapt to cystine limitation (termed “CLSA” cells) not only maintain glutathione synthesis despite low cystine levels but also undergo metabolic reprogramming. Key features of this reprogramming include a translational upregulation of the oxidative pentose phosphate pathway (OxPPP) (notably increased levels of G6PD, PGD, and PGLS), which fuels enhanced nucleotide synthesis and lipogenesis. The increased lipogenesis leads to higher triacylglyceride (TG) production and lipid droplet (LD) formation—a process that helps the cells mitigate lipotoxicity while promoting proliferation and ferroptosis resistance. A drug screen identified lomitapide, an FDA-approved lipid metabolism drug,

as a potent inhibitor of TG synthesis (likely via DGAT inhibition) in these adapted cells. Lomitapide not only induces lipotoxicity in CLSA cells but also sensitises PDAC cells and patient-derived models to chemotherapy (FOLFIRINOX), leading to improved anti-tumor efficacy in xenograft and genetically engineered mouse models. Overall, the study was well conducted and of broad interest. I only have a few remarks the authors might wish to address:

1) The authors report the activation of ATF4 at 24 hours upon cysteine limitation. Is the basal activation of ATF4 maintained in the CLSA cells? Expanding on this, does this generate a long-standing dependence on ATF4 activity, and ultimately, is the constitutive activation of ATF4 the causative event leading to increased dependency on TG synthesis? Finally, does the ATF4 activation also observed in the tumor samples? This could add molecular depth to some of the observations reported by the authors.

Response: As we shown in Fig. S1I, although ATF4 levels remained upregulated at up to 7 days after CLS exposure, its levels in CLSA cells return to levels comparable to those in control cells after 4 week adaptation. These data suggested that CLSA cells were able to overcome ISR despite marked reduction in intracellular cysteine. These observations are also consistent with the data showing global upregulation of mRNA

Figure R5, Representative IHC images (A) and H score quantitation (B) of ATF4 staining intensities in the PDAC tissues and para-tumor normal tissues from 60 PDAC patients.

translation, since ISR in general suppress global translation.

We performed IHC staining of ATF4 in PDAC tissues and paired paraneoplastic tissues. Our data showed that there was no difference in ATF4 staining between cancer and normal tissues (Figure R5). Taken together, our data suggested that ATF4 or ISR are unlikely to be responsible for CLSA phenotypes, including the dependency on TG synthesis.

2) The aspect describing the increase ferroptosis resistance of CLSA is not

worked out in detail. A better characterization, including evaluating the expression of GPX4, SLC7A11, FSP1, for example, would be of interest.

Response: We have previously investigated the effects of CLSA on multiple proteins involved in ferroptosis resistance (FSP1, COQ2, SLC7A11 etc) and these data are now included in this revised manuscript (Fig. S1O). In summary, we found that FSP1 was paradoxically downregulated in CLSA PDAC cells, but COQ2, the rate limiting enzyme in de novo CoQ10 synthesis, was upregulated. CLSA PDAC cells were also more sensitive to inhibitors targeting the CoQ10 pathway (iFSP1). SLC7A11 was downregulated in CLSA cells (Fig. S1I), but the cystine uptake activity was upregulated (Fig. S1J and S1K, please also see response to reviewer 1, comment 1). FSP1/DHODH inhibition or COQ2 knockdown could at least partially overcome ferroptosis resistance in CLSA cells (Fig. S1Q).

Along these lines, to what extent is the PUFA/MUFA ratio changed in the different phospholipid classes. Also, can the inhibition of LD restore the ferroptosis sensitivity in these cells? A recent preprint (<https://www.biorxiv.org/content/10.1101/2025.01.06.631537v1>) has hinted at a role of FSP1 in preserving TG and inhibiting ferroptosis within lipid droplets, perhaps this is worth discussing in line with the present observations.

Response: As suggested by this reviewer, we analyzed the C18:2/C18:1/C18:0 ratios, as well as the levels of C20:4, C22:4 and C22:6 in major lipid classes in naïve and CLSA PANC-1 cells and the new data are shown in Fig. S7. Overall, we found that ferroptosis-promoting PUFAs (C20:4, C22:4 and C22:6) were increased (instead of decreased) in major lipid classes. Therefore, the ferroptosis resistance in CLSA PDAC cells was likely not due to changes in PUFA levels in the lipidome.

We also discussed the bioRxiv preprint suggested by this reviewer and tested the effects of lomitapide, alone or in combination with RSL-3 in naïve and PDAC cells. Our new data showed that lomitapide treatment by itself didn't induce lipid peroxidation or ferroptosis (Fig. S5C, S5D). Lomitapide-induced cell death could be rescued by inhibitors for apoptosis and necroptosis, but not by ferrostatin-1 (Fig. S5C, S5D). However, when combined with RSL-3, lomitapide could sensitize PDAC cells (naïve or CLSA) to RSL3 and largely overcome CLSA-mediated ferroptosis resistance (Fig. S6E). These data are largely in line with recent evidence implicating a role for LD in ferroptosis resistance.

Minor points:

-Line 127 and Line 129, starting with “Although..” are duplicated.

-The word “reprogramming” is used in several places; the standard spelling is “reprogramming.”

-Line 121 - “ferroptosis-like cell death in PDAC cells in in 72 hours” contains a duplicate “in.” It should read “in 72 hours.”

-Line 448 - In the Discussion section, “miroenvironment” appears instead of “microenvironment.”

Response: These typos and mistakes have been corrected in the revised manuscript. Thank you!

Reviewer #3 (Remarks to the Author):

The article by Li et al. studies the adaptation to Cystine Limitation Stress (CLS) in PDAC. By using cell lines, patient-derived organoids, patient samples, and diverse mouse models, the authors demonstrate that adaptation to CLS promotes tumor growth by activating the oxidative pentose phosphate pathway via increased translation. CLSA induces lipid reprogramming and de novo lipid synthesis, resulting in triacylglyceride (TG) synthesis and accumulation in lipid droplets to mitigate lipotoxicity. The compound Lomitapide inhibits TG synthesis, interfering with lipid reprogramming and enhancing FOLFIRINOX chemotherapy both in vitro and in vivo.

I have thoroughly reviewed this manuscript and must commend the authors on the excellent quality of work presented. The research is insightful, well-structured, and contributes significantly to the field. The methodology is sound, and the conclusions drawn are both logical and impactful.

However, there are a few minor corrections and suggestions that I would like to point out:

1. Is LD staining increased in patients with high G6PD expression? These data would significantly strengthen the story, fully supporting the in vivo/in vitro data presented here on the connection between high oxPPP and increased de novo lipid synthesis.

Response: To address this question, we evaluated the levels of G6PD and LD levels in 30 patients. As shown in Fig. S8I-S8J, the expression levels of G6PD significantly correlated with LD levels, which is consistent with our in vitro data showing that activation of OxPPP could contribute to LD accumulation.

2. Considering that the patients used for survival analysis (Table S1) were treated with gemcitabine-based therapies, it would be important to determine if lomitapide sensitizes to gemcitabine or gemcitabine/abraxane on top of mFOLFIRINOX, at least using in vitro assays.

Response: As suggested by this reviewer, we tested the effect of lomitapide on PDAC cell sensitivities to Abraxane/Gemcitabine. Our data showed that lomitapide treatment

sensitized PDAC cells to Abraxane/Gemcitabine treatments in naïve and CLSA PDAC cells (Fig. S10D).

3. There are a few typographical errors throughout the text.

Response: The revised manuscript has been checked for typos and grammar mistakes.

4. The statistical analysis methodology must be justified and included in the Materials and Methods section.

Response: A statistical method section has been added.

- 1 Wang, X. *et al.* Mitochondrial Calcium Uniporter Drives Metastasis and Confers a Targetable Cystine Dependency in Pancreatic Cancer. *Cancer Res* **82**, 2254-2268 (2022). <https://doi.org/10.1158/0008-5472.CAN-21-3230>
- 2 D'Erasmus, L. *et al.* Efficacy and safety of lomitapide in homozygous familial hypercholesterolaemia: the pan-European retrospective observational study. *Eur J Prev Cardiol* **29**, 832-841 (2022). <https://doi.org/10.1093/eurjpc/zwab229>
- 3 Blom, D. J. *et al.* Long-Term Efficacy and Safety of the Microsomal Triglyceride Transfer Protein Inhibitor Lomitapide in Patients With Homozygous Familial Hypercholesterolemia. *Circulation* **136**, 332-335 (2017). <https://doi.org/10.1161/CIRCULATIONAHA.117.028208>
- 4 Cuchel, M., Blom, D. J. & Averna, M. R. Clinical experience of lomitapide therapy in patients with homozygous familial hypercholesterolaemia. *Atheroscler Suppl* **15**, 33-45 (2014). <https://doi.org/10.1016/j.atherosclerosissup.2014.07.005>
- 5 Yadav, B., Wennerberg, K., Aittokallio, T. & Tang, J. Searching for Drug Synergy in Complex Dose-Response Landscapes Using an Interaction Potency Model. *Comput Struct Biotechnol J* **13**, 504-513 (2015). <https://doi.org/10.1016/j.csbj.2015.09.001>

Point by point response

Reviewer #1 (Remarks to the Author):

I commend the authors for such a thorough and extensive revision!
Well done! In my opinion, the manuscript is now suitable for publication in Nature Communications

A few minor comments/suggestions:

1) I found both the BSA supplementation and the OA incorporation results very interesting and I definitely suggest to investigate these mechanisms in the future studies.

Response: We thank the reviewer for their thoughtful suggestions.

2) Lomitapide seems to be potent regardless of CLSA, so I would probably highlight this in the discussion

Response: The discussion has been modified according to the suggestion.

3) I spotted a few typos: lines 409 ("Taking" instead of "Taken"); 433 (Flg); 458, 576 ("Lipidomics" instead of "Lipidome")

4) There are 2 reference sections-all references should be together

5) Suppl Fig. 9 panel E misses the color legend

Response: All the mistakes have been corrected.

Reviewer #2 (Remarks to the Author):

The authors have sufficiently addressed my concerns.

I would only avoid stating that the resistance cells upregulate Coq biosynthesis proteins since the effect observed is only marginal.

Response: We have changed the wording "increase" to "moderately increase" following the suggestion of this reviewer.